  

# TBC1D14 regulates autophagy via the TRAPP complex and ATG9 traffic

Christopher A Lamb[1], Stefanie Nühlen[2], Delphine Judith[1], David Frith[3], Ambrosius P Snijders[3], Christian Behrends[2] & Sharon A Tooze[1,*]

## Abstract

Macroautophagy requires membrane trafficking and remodelling to form the autophagosome and deliver its contents to lysosomes for degradation. We have previously identified the TBC domain-containing protein, TBC1D14, as a negative regulator of autophagy that controls delivery of membranes from RAB11-positive recycling endosomes to forming autophagosomes. In this study, we identify the TRAPP complex, a multi-subunit tethering complex and GEF for RAB1, as an interactor of TBC1D14. TBC1D14 binds to the TRAPP complex via an N-terminal 103 amino acid region, and overexpression of this region inhibits both autophagy and secretory traffic. TRAPPC8, the mammalian orthologue of a yeast autophagy-specific TRAPP subunit, forms part of a mammalian TRAPPIII-like complex and both this complex and TBC1D14 are needed for RAB1 activation. TRAPPC8 modulates autophagy and secretory trafficking and is required for TBC1D14 to bind TRAPPIII. Importantly, TBC1D14 and TRAPPIII regulate ATG9 trafficking independently of ULK1. We propose a model whereby TBC1D14 and TRAPPIII regulate a constitutive trafficking step from peripheral recycling endosomes to the early Golgi, maintaining the cycling pool of ATG9 required for initiation of autophagy.

**Keywords** Autophagy; Membrane Trafficking; Rab proteins; TRAPP
**Subject Categories** Autophagy & Cell Death; Membrane & Intracellular Transport
The EMBO Journal (2016) 35: 281–301

## Introduction

To remain healthy, cells must clear their cytosol of misfolded proteins, dysfunctional organelles and intracellular pathogens. To this end, eukaryotes employ the evolutionarily conserved autophagy pathways (Mizushima *et al*, 2008). Macroautophagy (here referred to as autophagy) is a highly dynamic process involving the formation of a double-membrane cisterna (the phagophore or isolation membrane), which expands to engulf portions of the cytosol, closing to form an autophagosome (Lamb *et al*, 2013b). Through the use of receptors, of which p62 is prototypical, autophagosomes can also be directed to engulf specific cargos (Birgisdottir *et al*, 2013). The completed autophagosome then matures by fusing with the lysosome, allowing the contents of the autophagosome to be degraded and returned to the cytosol. This recycling function is crucial for cells and organisms to survive periods of stress, such as amino acid starvation, growth factor withdrawal and hypoxia (Lamb *et al*, 2013b), and dysregulation of the autophagy pathways plays a role in pathological states including ageing, bacterial infection, neurodegeneration and cancer (Mizushima *et al*, 2008).

The core autophagy machinery and the genes encoding it were originally characterised in *Saccharomyces cerevisiae* (yeast) through genetic screening. There are currently 40 autophagy-related (ATG) genes known in yeast, many of which have mammalian orthologues, and the conserved core Atg proteins fall into several groups. Upon amino acid withdrawal, the mammalian target of rapamycin complex 1 (mTORC1) is inactivated, which removes repression on the ULK (uncoordinated 51-like kinase) complex, which consists of ULK1/2, ATG13, FIP200 and ATG101(Hara *et al*, 2008; Chan *et al*, 2009; Hosokawa *et al*, 2009; Mercer *et al*, 2009). The ULK1 complex then goes on to activate the autophagy-specific phosphatidylinositol 3 kinase (PtdIns(3)K) complex, which includes ATG14, Beclin1, VPS34 and p150 and nucleates pools of phosphatidylinositol-3-phosphate (PtdIns(3)P) at specific sites called omegasomes on the endoplasmic reticulum (ER) marked by double FYVE domain-containing protein 1 (DFCP1), where the ER is thought to act as a cradle for autophagosome formation (Axe *et al*, 2008; Hayashi-Nishino *et al*, 2009; Yla-Anttila *et al*, 2009). The phagophore, a double lipid bilayer structure, is formed from the omegasome and expands through the action of two ubiquitination-like enzymatic cascades. The first of these involves ATG7 (E1) and ATG10 (E2) and results in conjugation of the ubiquitin-like modifier ATG12 to ATG5 (Mizushima *et al*, 1998). The ATG12-5 complex associates with ATG16, acting as the E3 enzyme in the second cascade with ATG7 (E1) and ATG3 (E2), and this complex supports covalent attachment of ATG8 family members (LC3 and GABARAP proteins in

1 Molecular Cell Biology of Autophagy Group, Francis Crick Institute, London, UK
2 Institute of Biochemistry II, Medical School Goethe University, Frankfurt, Germany
3 The Francis Crick Institute Mass Spectrometry Core Technology Platform, Clare Hall Laboratories, Potters Bar,UK
 *Corresponding author. Tel: +44 207 269 3122; E-mail: Sharon.tooze@crick.ac.uk

mammals) to the lipid phosphatidylethanolamine (PE) at the phagophore (Mizushima et al, 1998, 2003), dependent on the PtdIns(3)P-binding protein WIPI2 (Dooley et al, 2014). Lipidated ATG8 proteins associate with the phagophore as it expands and closes to form the autophagosome, and remain one of the key markers for autophagosome formation (Kabeya et al, 2004; Klionsky et al, 2012).

Of note, ATG9 is the only transmembrane protein required for progression of autophagy (Webber & Tooze, 2010). Yeast Atg9 has been found to localise to small cytoplasmic vesicles, several of which appear to nucleate the PAS (pre-autophagosomal structure; a single structure in yeast cells thought to be functionally equivalent to mammalian omegasomes) under starvation conditions, eventually becoming incorporated into the growing phagophore (Yamamoto et al, 2012). Importantly, Atg9 does not persist in the autophagosomal membrane after fusion with the vacuole—it is recycled back to the cytoplasmic vesicles to participate in further rounds of PAS formation (Yamamoto et al, 2012). Atg9 traffic is under the control of Atg1 kinase, which phosphorylates Atg9 directly to control phagophore elongation (Papinski et al, 2014).

The Atg9 vesicles are directed to the PAS by the small GTPase Ypt1 and its GEF (GTP exchange factor) the TRAPPIII (trafficking protein particle III) complex (Kakuta et al, 2012); a large, multisubunit tethering complex conserved from yeast to humans (Sacher et al, 1998; Barrowman et al, 2010). In yeast, the TRAPPs are organised into three complexes—the core TRAPPI complex, involved in ER-Golgi traffic, the larger TRAPPII complex, which plays a role in endosome to Golgi and intra-Golgi traffic, and the TRAPPIII complex, which is essentially the TRAPPI complex with an additional subunit, Trs85 (Barrowman et al, 2010). TRAPPIII and Ypt1 activities have been linked to autophagy (Nazarko et al, 2005; Lynch-Day et al, 2010; Kakuta et al, 2012; Lipatova et al, 2012; Shirahama-Noda et al, 2013).

As in yeast, the trafficking of ATG9 appears to be under the control of ULK1 (an orthologue of Atg1) in mammals (Young et al, 2006) and is crucial for the early stages of autophagy, as fewer omegasomes form in cells depleted for ATG9 (Winslow et al, 2010; Orsi et al, 2012). In contrast to yeast, however, we showed that ATG9 vesicles do not fully fuse with the growing phagophore; rather, they transiently interact with both GFP-DFCP1- and GFP-LC3-positive structures suggesting that ATG9-positive vesicles play a role throughout autophagosome formation (Orsi et al, 2012).

RAB1 (the mammalian orthologue of Ypt1) is known to regulate formation of DFCP1-positive structures and autophagosomes (Winslow et al, 2010; Zoppino et al, 2010; Mochizuki et al, 2013) and association of LC3B with Salmonella (Huang et al, 2011). Mammalian TRAPP subunits have been linked to autophagy in a large-scale proteomics study (Behrends et al, 2010) and knockdown of a core TRAPP subunit reduces the association of LC3B with intracellular Salmonella (Huang et al, 2011), although unlike yeast, no TRAPP-dependent regulation of ATG9 trafficking has yet been identified. However, orthologues of most yeast TRAPP subunits are present in mammals (Scrivens et al, 2011) and recent data generated using epitope-tagged proteins indicates that two different TRAPP complexes may exist in mammalian cells, broadly similar to yeast TRAPPII and TRAPPIII with additional metazoan-specific subunits (Bassik et al, 2013).

Aside from ATG9 membranes, numerous other sources provide membranes for the growing phagophore. It receives input from the ER-Golgi intermediate compartment (Ge et al, 2013), Golgi (Young et al, 2006; Itoh et al, 2008), recycling endosomes (Longatti et al, 2012; Knaevelsrud et al, 2013; Puri et al, 2013) and the plasma membrane (PM; Ravikumar et al, 2010; Moreau et al, 2011, 2012). As completion of the autophagosome requires expansion into large membrane vesicles and fusion with other endomembrane compartments to permit degradation of autophagy cargos, the vesicle trafficking machinery of the cell is vital for its progression. Indeed, a growing list of membrane trafficking regulators impinges on autophagosome formation and maturation. These have been reviewed in detail recently (Lamb et al, 2013a,b) and include small GTPases (Itoh et al, 2008; Zoppino et al, 2010; Moreau et al, 2012), RabGAPs (GTPase activating proteins) (Itoh et al, 2011; Longatti et al, 2012; Popovic et al, 2012), SNARE proteins (Itakura et al, 2012; Hamasaki et al, 2013; Puri et al, 2013; Moreau et al, 2014), sorting nexins (Knaevelsrud et al, 2013) and vesicle tethering complexes (Liang et al, 2008). Unpicking the functions of these components will permit a better understanding of the mechanisms of autophagosome formation.

We previously carried out an overexpression screen of mammalian TBC (Tre-Bub-CDC16) domain-containing RABGAPs to identify those involved in autophagy (Longatti et al, 2012). We identified ten TBC proteins that reduced LC3 lipidation on overexpression—several of which were subsequently found to interact with LC3 family proteins via LIR (LC3 interacting region) motifs (Popovic et al, 2012). We focused on TBC1D14, a previously rather poorly characterised TBC protein (Haas et al, 2007; Tempel et al, 2008; Ching et al, 2010) due to its membrane trafficking phenotype and co-localisation with core autophagy proteins. TBC1D14 overexpression resulted in formation of an enlarged, ULK1-positive tubulated recycling endosome (RE) compartment, inhibiting autophagy (Longatti et al, 2012). REs were found to contribute membranes to autophagosomes and also harbour several key autophagy regulators including ATG9 and the ULK complex, findings which have subsequently been confirmed by other groups (Knaevelsrud et al, 2013; Puri et al, 2013). In particular, one study revealed that SNX18 positively regulates autophagosome formation, and also generates RE tubules (Knaevelsrud et al, 2013). Despite TBC1D14 being able to bind to RAB11, it appears not to act as a GAP for RAB11 and may act as an effector (Longatti et al, 2012), so how TBC1D14 affected recycling endosome morphology and function in autophagy remained unclear.

In the present study, we sought to understand how TBC1D14 functions in vesicle traffic and autophagy by identifying candidate interactors. This approach identified the TRAPP complex as an interacting partner of TBC1D14. We have characterised the interaction between TRAPP and TBC1D14 and found that the dominant negative effect of overexpressed TBC1D14 we previously identified is due to its binding to a TRAPP complex. We show that TRAPPC8, the mammalian orthologue of the yeast autophagy-specific TRAPP subunit Trs85, mediates the interaction between TRAPP and TBC1D14, and provide evidence at the endogenous level of the existence of a TRAPPIII-like complex in mammalian cells. We also demonstrate that TBC1D14 is required for activation of RAB1 through its TRAPP-binding ability. Finally, we have shown that disruption of TRAPP function results in defects in secretory traffic and autophagosome formation, and the autophagy defect results from dysregulation of ATG9 traffic.

# Results

### Identification of the TRAPP complex as a TBC1D14 interactor

To identify TBC1D14 interactors, a GST pull-down assay was performed on lysate from HEK293A cells using immobilised GST or GST-TBC1D14. Candidate bands were excised and analysed by mass spectrometry. The analysis yielded several members of the TRAPP (trafficking protein particle) complex (Fig 1A, Appendix Table S1).

Using an immunoprecipitation (IP) approach, we validated that GFP-TBC1D14 could co-precipitate TRAPP subunits including TRAPPC4 (a conserved "core" subunit) and TRAPPC12/TTC15 (a metazoan-specific TRAPP subunit; Scrivens *et al*, 2011), whereas GFP alone could not (Fig 1B). Using a GFP-tagged TRAPP subunit, GFP-TRAPPC3 (also a conserved "core" subunit; Barrowman *et al*, 2010), we found that endogenous TBC1D14 could be precipitated by the TRAPP complex (Fig 1C).

As TBC1D14 overexpression generates an exaggerated, tubulated transferrin-positive RE compartment (Longatti *et al*, 2012) and the TRAPP complex interacts with TBC1D14, we tested whether TRAPP subunits were present on the transferrin-positive tubules using confocal microscopy. Indeed, both endogenous TRAPPC4 and TRAPPC12 co-localised with the GFP-TBC1D14 and Alexa-647 trans-ferrin-positive tubulated endosomes in HEK293A cells (Fig 1D and E). We have previously demonstrated that endogenous TBC1D14 is localised to the Golgi stack (Longatti *et al*, 2012), and we observed that the core TRAPP subunit TRAPPC4 and its GEF target RAB1B both partially localise to the Golgi with TBC1D14 in HEK293A cells (Fig 1F). These data suggest that the TRAPP complex, TBC1D14 and RAB1 may function together at the Golgi, and the localisation of TBC1D14 to tubulated RE evident after its overexpression also results in mis-localisation of the TRAPP complex.

As we have previously demonstrated, the tubulated ERC (endo-cytic recycling compartment) generated by TBC1D14 overexpression is positive for endogenous RAB11, and indeed, RAB11 is required for the tubules to form (Longatti *et al*, 2012). We decided to investigate whether the tubules can simultaneously harbour RAB11 and RAB1, which under control conditions are largely on separate compartments (Fig 2A, upper panels). The TBC1D14 tubules are positive for RAB11 as expected, and a subset of the tubules harbour RAB1B (Fig 2A, lower panels and inset).

Previous studies have shown that membrane traffic and potentially membrane contact sites can occur between RAB11- and RAB1-positive juxtanuclear compartments (Marie *et al*, 2009). We found that endogenous RAB1 and RAB11 co-localised on stable vesicular structures which are resistant to Brefeldin A (BFA), an inhibitor of Sec7 domain-containing proteins which fragments the Golgi, confirming the results of Marie *et al*, 2009;. Importantly, these structures do not co-localise with the early Golgi marker GM130, which localises to ER exit sites in BFA-treated cells (Mardones *et al*, 2006; Fig EV1). These structures represent stable trafficking intermediates between RAB11-positive recycling endosomes and RAB1-positive transport vesicles.

Given that TBC1D14 can bind TRAPP on the GFP-TBC1D14-induced tubules, we tested whether RAB1 or RAB11 co-localised with the core TRAPP subunit TRAPPC4. TRAPPC4 and RAB11A co-localised extensively on the tubules (Fig 2B). Similarly, TRAPPC4 and RAB1B also co-localised on the GFP-TBC1D14-induced tubules (Fig 2C). These data support a model where TBC1D14 acts as a bridge between a TRAPP complex and activated RAB11. This membrane-localised TRAPP can then act as a GEF for RAB1, recruiting it to membranes (Fig 2D).

### Amino acids 120–223 of TBC1D14 are required for TRAPP complex interaction

Having confirmed the mass spectrometry interactions and observed the striking accumulation of TRAPP on TBC1D14-positive membranes, we determined which region of TBC1D14 was responsible for TRAPP binding. TBC1D14 has a typical domain structure for TBC domain-containing RabGAPs (Frasa *et al*, 2012), with the TBC domain at the C-terminus of the protein (Fig 3A). Although the structure of the TBC domain has been solved (residues 357–672; Tempel *et al*, 2008), nothing is known about the function of the N-terminal part of the protein. N-terminally GFP-tagged TBC1D14 constructs were generated and transfected into HEK293A cells, and the cell lysates were subject to GFP-Trap IP and blotting for TRAPPC12 and TRAPPC4. We show that the C-terminus of the protein (residues 224–669, encompassing the putative TBC domain from residues 411–611, Fig 3A) does not bind to the tested TRAPP subunits; however, a 103 amino acid stretch of TBC1D14 (residues 120–223) is sufficient for TRAPP interaction (Fig 3B). This is distinct from the region (residues 224–330) we have previously shown to be involved in an interaction with ULK1 kinase (Longatti *et al*, 2012). TBC1D14 residues 120–223 will subsequently be referred to as the TRAPP-binding region (TBR).

---

**Figure 1.  TBC1D14 interacts with the TRAPP complex.**

A  GST pulldown used to analyse TBC1D14 interactors. Recombinant GST or GST-TBC1D14 was incubated with or without HEK293A lysate. Bound proteins were eluted from the beads using Laemmli sample buffer and subjected to SDS–PAGE, with visible bands being excised from the gel and analysed by mass spectrometry.

B  Lysates of HEK293A cells expressing either GFP or GFP-TBC1D14 were subjected to immunoprecipitation (IP) with GFP-Trap and then immunoblotted for GFP, TRAPPC12 and TRAPPC4. White arrowhead indicates GFP, black arrowhead indicates GFP-TBC1D14.

C  Lysates of cells expressing GFP or GFP-TRAPPC3 were subjected to IP with GFP-Trap and then immunoblotted for TRAPPC4, GFP and TBC1D14. White arrowhead indicates GFP, black arrowhead indicates GFP-TRAPPC3.

D  Cells expressing GFP-TBC1D14, generating a tubulated endosomal compartment, were fed Alexa-647-labelled transferrin (white, blue in merge) for 15 min in full medium, fixed, stained for endogenous TRAPPC4 (red) and analysed by confocal microscopy.

E  Cells treated as in (D) were stained for endogenous TRAPPC12 (red) and analysed by confocal microscopy.

F  Cells in complete medium were fixed, stained for endogenous TBC1D14 (green), RAB1B (red) and TRAPPC4 (white, blue in merge) and analysed by confocal microscopy. Inset shows protein localisation in juxtanuclear area.

Data information: In (D–F), white arrowheads indicate transfected cells, and yellow arrowheads depict regions of co-localisation in the inset. Scale bars, 10 μm. Data shown in (B–F) are representative of 3 independent experiments.

## TBR overexpression affects the Golgi and the secretory pathway, not recycling endosome function

Although we have previously shown that full-length TBC1D14 modulates autophagy by altering vesicle traffic through recycling

endosomes (Longatti *et al*, 2012), our data suggest that the N-terminal TBR may modulate membrane traffic through its TRAPP-binding capacity. We first tested whether GFP-TBR can generate the tubular RE phenotype seen on overexpression of TBC1D14 (Fig 4A). The TBR alone appears to be cytosolic and does not generate the

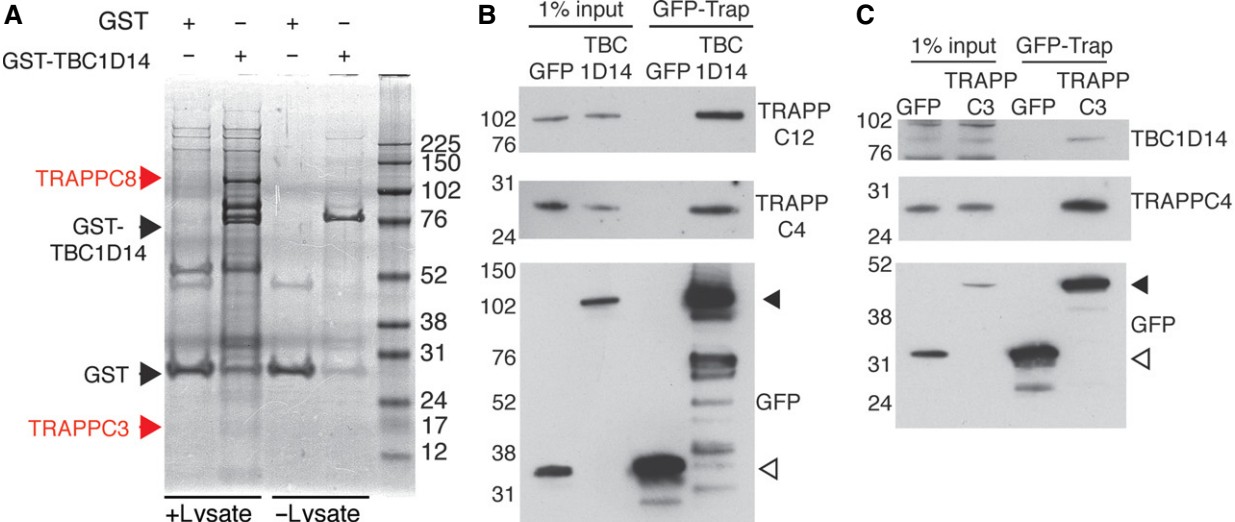

**Figure 1.**

**Figure 2.  A subset of GFP-TBC1D14-induced tubules harbour RAB1.**

A   HEK293A cells transfected with GFP (upper panels) or GFP-TBC1D14 (lower panels) (green) were labelled with anti-RAB11A (red) and anti-RAB1B (white, blue in merge) antibodies. White arrows depict transfected cells. Inset: GFP-TBC1D14-transfected cell showing triple co-localisation between GFP-TBC1D14, RAB1A and RAB1B (white arrow heads) and double co-localisation between GFP-TBC1D14 and RAB11A (yellow arrowheads).

B   Co-localisation of GFP-TBC1D14, TRAPPC4 (red) and RAB11A (white, blue in merge). White arrowheads indicate regions of triple co-localisation.

C   Co-localisation of GFP-TBC1D14, TRAPPC4 (red) and RAB1B (white, blue in merge). White arrowheads indicate regions of triple co-localisation and yellow arrowheads regions of double co-localisation between GFP-TBC1D14 and TRAPPC4.

D   Model for recruitment of RAB1 to tubulated RE. RAB11-GTP recruits TBC1D14 to the ERC membrane, and TBC1D14 in turn recruits TRAPP. TRAPP acts as a GEF to recruit RAB1 aberrantly to ERC membranes.

Data information: Scale bars, 20 μm.

tubulated RE phenotype—in many transfected HEK293A cells, the morphology of the RE compartment appears to be intact (Fig 4A). However, the C-terminal region of the protein (224–669), encompassing the TBC domain, is membrane associated and generates an enlarged juxtanuclear transferrin-positive structure with peripheral puncta positive for GFP-224–669 and transferrin (Fig 4A).

To quantitatively assay the effect on RE function, we utilised a pulse-chase approach where cells transfected with GFP-TBC1D14 and deletion mutants were fed Alexa-647 transferrin (Tfn) for 15 min in full medium, and the fluorescent Tfn chased out of the cells at 15-min intervals. The GFP-positive cells were sorted and their Tfn content analysed by flow cytometry. This showed that while expression of GFP-TBC1D14 or GFP-224-669 impairs Tfn recycling, GFP-TBR has no effect on Tfn recycling when compared to GFP alone (Fig 4B).

As TBR expression does not affect RE function, what other membrane trafficking pathways could a TBC1D14–TRAPP interaction be impacting that may modulate autophagy? A large-scale study of putative RabGAPs required for normal Golgi morphology showed that TBC1D14 overexpression resulted in fragmented Golgi stacks in two different cell lines (Haas *et al*, 2007). We confirmed this phenotype using the established *cis*-Golgi marker GM130 (Fig 4C, compare GFP to GFP-TBC1D14). Remarkably, overexpression of GFP-TBR also fragmented the Golgi (Fig 4C, GFP-TBR), whereas overexpression of GFP-224-669 did not (Fig 4C, GFP-224–669). Importantly, previous studies have shown that mammalian TRAPP subunits are needed for maintenance of Golgi structure (Yamasaki *et al*, 2009; Scrivens *et al*, 2011), supporting the idea that the TRAPP complex and TBC1D14 may be acting in the same pathway.

We used the HeLa C1 cell line (Gordon *et al*, 2010) to assay the effect TBR-induced Golgi fragmentation had on constitutive secretion. These cells stably express a GFP-tagged reporter construct (GFP-FM4-hGH), which accumulates in the ER due to oligomerisation of its tandem FKBP repeats. On treatment with the rapamycin analogue AP21998 (D/D solubiliser, Clontech), the oligomers are disrupted and the reporter travels through the constitutive secretory pathway and is secreted; thus, the total amount of intracellular GFP decreases over time (Gordon *et al*, 2010). This means that the change in GFP fluorescence over time can be used to measure constitutive secretion.

C1 cells expressing CFP or CFP-TBR were treated with D/D solubiliser over an 80-min time course, and their GFP content analysed by flow cytometry (Fig 4D). As a control, we also treated C1 cells with BFA, which blocks constitutive secretion in this and other cell models (Rosa *et al*, 1992; Gordon *et al*, 2010). The results show that expression of CFP-TBR reduces secretion of GFP-FM4-hGH. This finding, in conjunction with the Golgi fragmentation data, shows

that TBR overexpression alters Golgi function. Therefore, TBC1D14's function impinges on both the secretory and endocytic pathways likely through its N- and C-terminal domains, respectively. These data further support the notion that TBC1D14 binds the TRAPP complex, as TRAPP subunits are involved in secretion (Barrowman *et al*, 2010).

As the D/D solubiliser is a rapamycin analogue, we tested whether mTOR was inactivated in C1 cells by the D/D treatment, using as controls the known mTOR inhibitors rapamycin (Sabatini *et al*, 1994) and Torin 1 (Liu *et al*, 2010). After an 80-min incubation, we immunoblotted for GFP, to monitor GFP-FM4-hGH secretion, and the phosphorylated form of the S6 ribosomal protein, a downstream target of the mTOR-activated p70 S6-kinase (Chung *et al*, 1992). Our data show that treatment of cells with rapamycin or D/D solubiliser promotes secretion of the GFP-FM4-hGH reporter construct; this is not the case with Torin treatment (Appendix Fig S1A). Moreover, S6 phosphorylation was not inhibited by D/D solubiliser treatment but was inhibited by rapamycin and Torin treatment (Appendix Fig S1A). We used the same treatments on HEK293A cells and analysed lipidation of LC3-I to LC3-II, a standard read-out for autophagic flux (Kabeya *et al*, 2000), and found that unlike rapamycin and Torin treatment, D/D solubiliser treatment does not induce lipidation of LC3-I to LC3-II (Appendix Fig S1B), showing that the D/D solubiliser does not promote autophagosome formation.

### Overexpression of the TBR modulates autophagy at an early step

We have previously demonstrated that TBC1D14 overexpression has an inhibitory effect on autophagy as measured by LC3 lipidation (Longatti *et al*, 2012), via its modulation of vesicle traffic through the RE. While part of this inhibition is likely due to TBC1D14's RAB11- and ULK1-binding functions, based on TRAPP's autophagic function in yeast, we decided to test whether overexpression of TBR alone could negatively regulate autophagy. HEK293A cells expressing GFP-TBR displayed a reduction in LC3 lipidation on starvation with EBSS in the presence of bafilomycin A1 (BafA1), when compared to control conditions, suggesting that autophagosome formation is impaired (Fig 5A). In contrast, cells expressing GFP-224–669 showed a small increase in LC3 lipidation on starvation suggesting that the inhibition of autophagy observed upon TBC1D14 overexpression (Longatti *et al*, 2012) is mainly due to the mis-localisation of the TRAPP complex to the tubulated RE (Fig 1D and E).

In order to test at which point in the autophagy pathway TBR overexpression had its effect, we used several fluorescence markers for autophagosome formation. Firstly, we used HEK293A GFP-LC3B cells (2GL9, (Chan *et al*, 2007)) and determined the number of GFP-LC3B

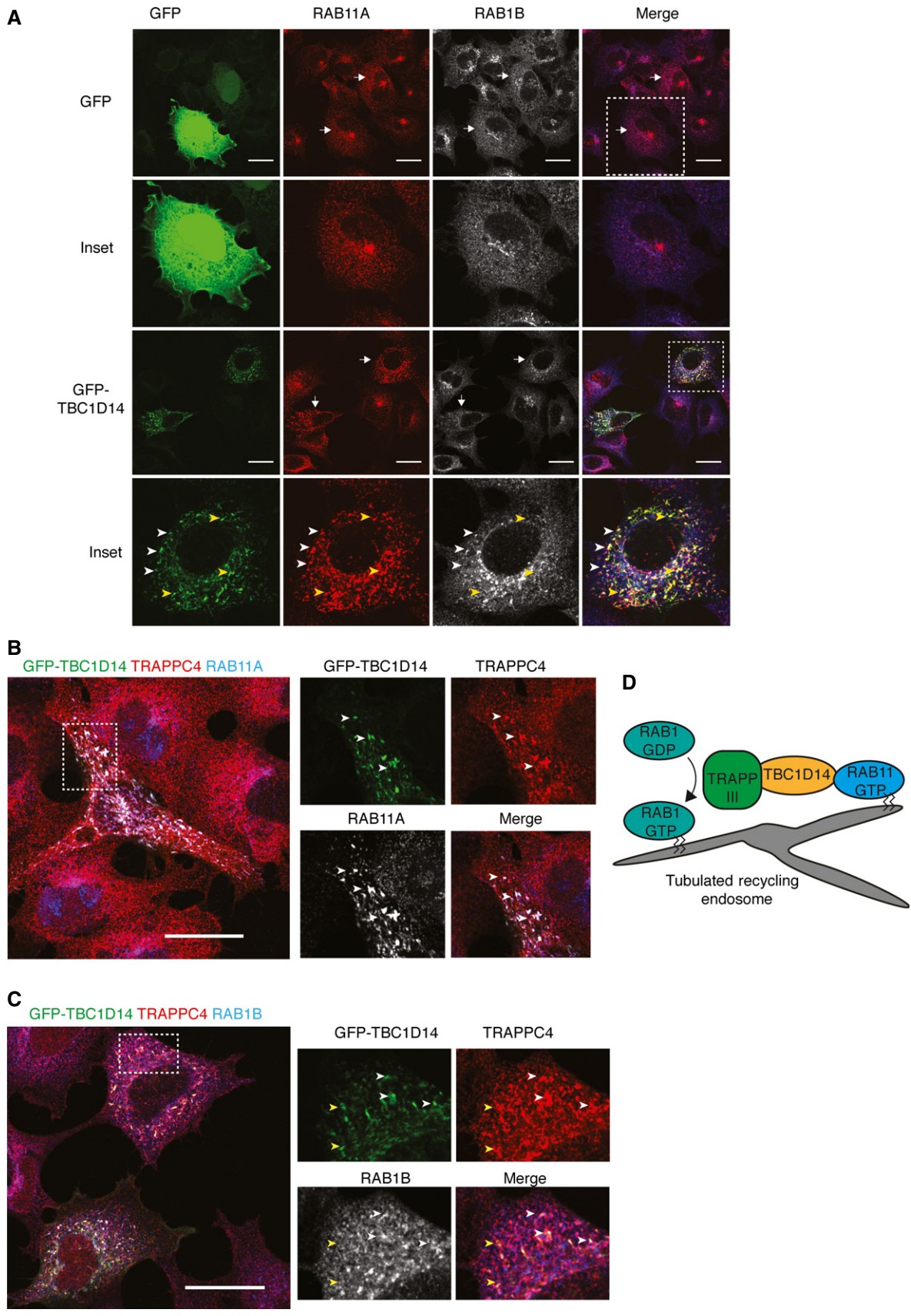

**Figure 2.**

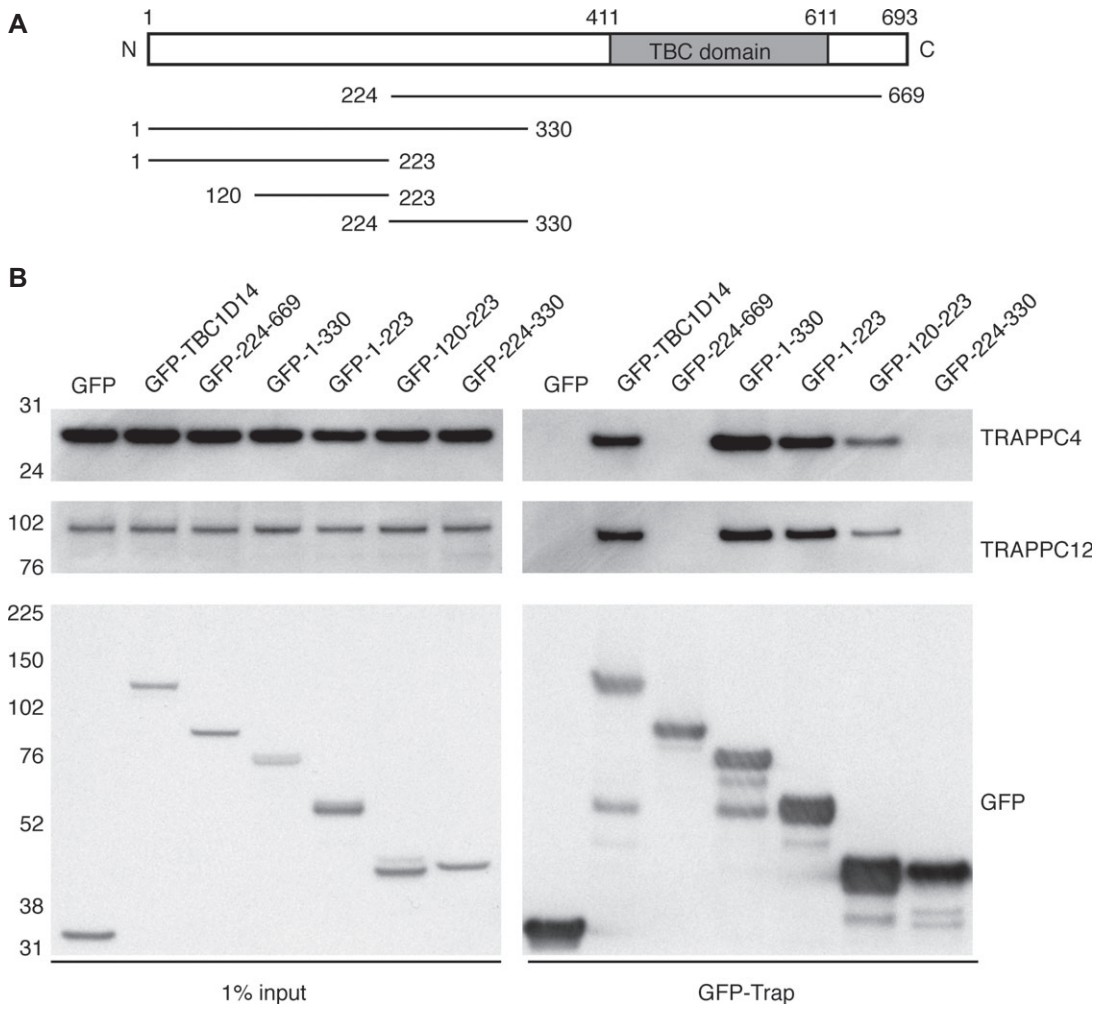

**Figure 3.  TBC1D14 binds the TRAPP complex via a 103 amino acid N-terminal region.**

A  Schematic diagram of full-length TBC1D14, indicating the C-terminal TBC domain (shaded), and truncation mutants used in subsequent analyses.
B  HEK293A cell lysates (from approximately $8 \times 10^6$ cells per construct) expressing the indicated GFP fusions were subjected to IP and immunoblotting for GFP, TRAPPC12 and TRAPPC4. Data are representative of 3 independent experiments.

puncta formed upon starvation. As expected from the lipidation data, fewer LC3B puncta formed on expression of mCherry-TBR compared to mCherry alone (Fig 5B). As LC3B lipidation and membrane association is a later step in the autophagosome formation pathway, we also looked at the formation of WIPI2 and GFP-DFCP1 puncta upon starvation. These proteins bind PtdIns(3)P, a marker for omegasomes and

autophagosomes which represent early steps in autophagosome formation (Axe *et al*, 2008; Polson *et al*, 2010; Dooley *et al*, 2014). In both cases, we found a small but significant reduction in the number of WIPI2 and GFP-DFCP1 puncta upon starvation (Fig 5C and D). In summary, these data indicate that TBR overexpression inhibits an early stage in autophagosome formation.

**Figure 4.  Expression of the TRAPP-binding region of TBC1D14 impairs Golgi function.**  ▶

A  Cells expressing the indicated GFP fusions (green) were loaded with Alexa-647 transferrin (red; white arrowheads indicate transfected cells) for 15 min, fixed and the morphology of the transferrin-positive recycling endosomes analysed by confocal microscopy. Scale bars, 10 μm. Arrows in inset indicate the juxtanuclear ERC.
B  Cells transfected as in (A) were loaded with Alexa-647 transferrin for 15 min and the fluorescent transferrin chased out for the indicated time periods. At the end of the time course, the cells were fixed and the transferrin content of GFP-positive cells analysed by flow cytometry. Results are expressed as a percentage of Tfn fluorescence at $t = 0$ and are the mean of 3 independent experiments, ± s.e.m.
C  Cells transfected as in (A) were stained for the *cis*-Golgi marker GM130. White arrowheads depict transfected cells. Scale bars, 10 μm. Bar chart is quantification of fragmented Golgi complex (> 100 cells per condition from three independent experiments, ± s.e.m. **$P < 0.01$, one-way ANOVA with Sidak's post-test).
D  HeLa C1 cells transfected with CFP or CFP-TBR were treated with D/D solubiliser (Clontech) or D/D solubiliser plus 2.5 μg/ml Brefeldin A (BFA) for the indicated time periods, trypsinised, fixed and their GFP content analysed by flow cytometry. Results are expressed as percentage of GFP levels at $t = 0$, and the mean of 3 independent experiments ± s.e.m. is shown.

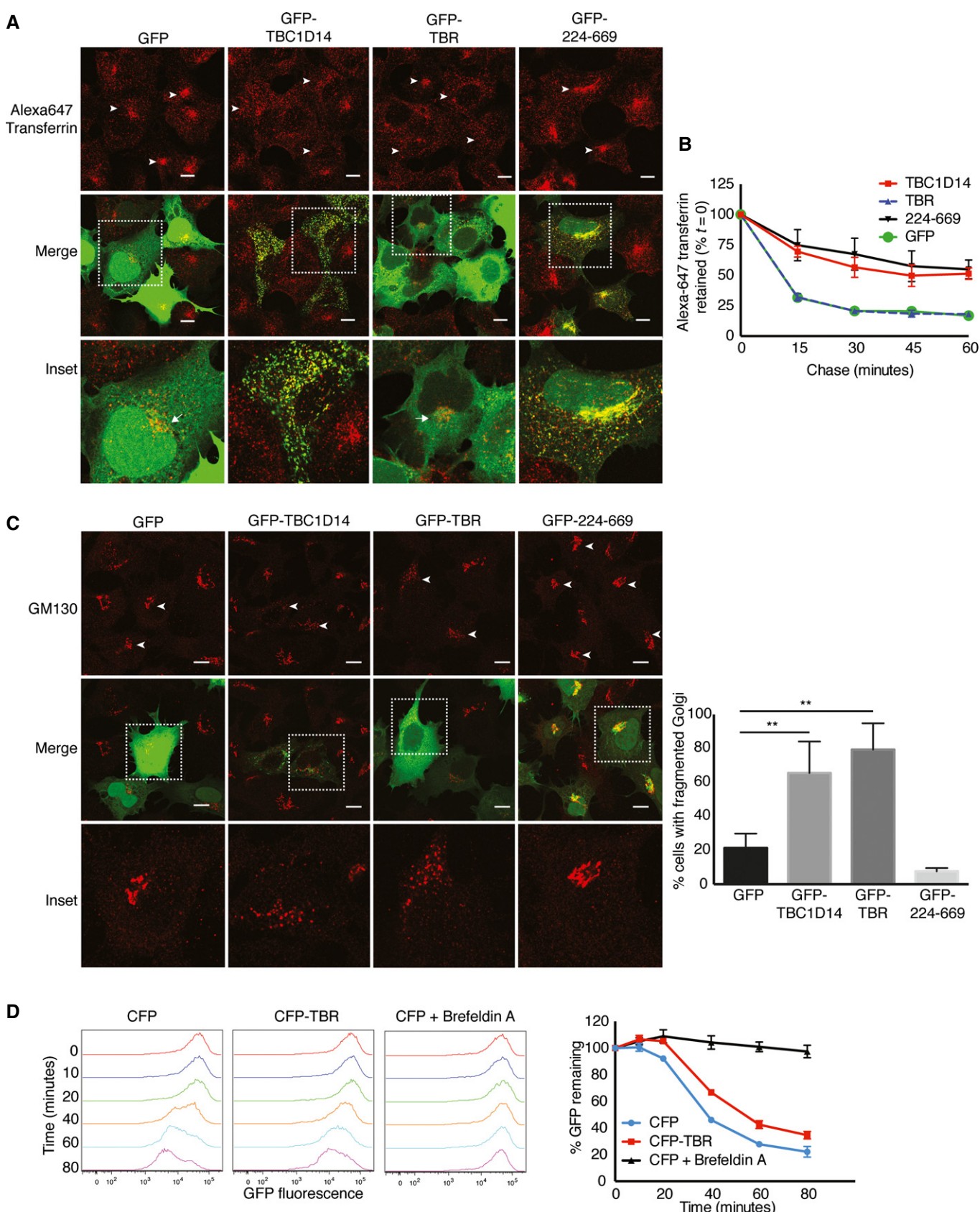

**Figure 4**

## TRAPPC8 is the most proximal TRAPP subunit to TBC1D14 and mediates the interaction between TBR and the core TRAPP complex

The TRAPP complex is a large, multi-subunit complex, with three distinct variants (I, II and III) in yeast cells (Barrowman *et al*, 2010). In mammals, despite the presence of orthologues of most of the yeast TRAPP subunits (Scrivens *et al*, 2011), the existence of multiple TRAPP complexes at the endogenous level is unclear. We sought to identify which of the TRAPP subunits are most proximal to TBC1D14 and may mediate the interaction between TBC1D14 and TRAPP. We adapted the BioID system (Roux *et al*, 2012) combined with mass spectrometry to study this interaction (Fig 6A). Briefly, cells were transfected with myc-BioID alone or myc-BioID-TBC1D14 and grown overnight in media supplemented with 50 μM biotin. The cells were lysed, the samples split, and half of the sample denatured with 1% SDS. The denatured and non-denatured samples were incubated with streptavidin resin, and then bound proteins were eluted, subject to SDS–PAGE and gel lanes analysed by mass spectrometry. Proteins that are biotinylated and thus bind to streptavidin resin under denaturing conditions are presumed to be most proximal to the myc-BioID tagged construct.

The scatterplots in Fig EV2A and B show label-free quantification of TRAPP subunits pulled down in the BioID experiment using log$_{10}$-transformed intensity-based absolute quantification (iBAQ) values calculated from mass spectrometry data (Schwanhausser *et al*, 2011). The TRAPP subunits are pulled down by myc-BioID-TBC1D14 under non-denaturing conditions, compared to myc-BioID alone (Fig EV2A). These data confirm that we can isolate TRAPP subunits by this method as with the IP approaches in Fig 1. Fig EV2B compares TRAPP subunits pulled down by myc-BioID-TBC1D14 in denaturing versus non-denaturing conditions. Approximately equal amounts of TBC1D14 are purified in both conditions indicating that the biotin–streptavidin interaction is not disrupted under the 1% SDS conditions. This was also the case for TRAPPC8, to a lesser extent TRAPPC2L and TRAPPC10 but not for other TRAPP subunits.

The proteins bound to the streptavidin resin in the myc-BioID-TBC1D14 denaturing pulldown are likely to be most proximal to TBC1D14 as they have been biotinylated. The results indicate that TRAPPC8 is the most proximal subunit to TBC1D14 (the proposed biotinylation range of myc-BioID is < 20 nm (Roux *et al*, 2012)) as it has high iBAQ scores in both denaturing and non-denaturing conditions, behaving similarly to TBC1D14.

We confirmed biotinylation of TRAPPC8 under denaturing conditions by Western blot analysis of the streptavidin pulldown (Fig 6B). TRAPPC4 and TRAPPC12, subunits predicted to be less

proximal by the BioID mass spectrometry analysis, were not found to be biotinylated in the Western blot analysis, supporting the mass spectrometry data.

These data indicate that TRAPPC8 may be the most proximal subunit to TBC1D14, and we hypothesised that TRAPPC8 mediates interaction of TBC1D14 with the rest of the TRAPP complex. To test this, we knocked down TRAPPC8 in HEK293A cells and incubated the lysates with immobilised GST-TBR. This showed that depletion of TRAPPC8 prevented interaction of the core TRAPP subunit TRAPPC4 with the TBR (Fig 6C). This loss of TRAPPC4 binding does not appear to be due to the core TRAPP complex failing to assemble normally in the absence of TRAPPC8, as GFP-TRAPPC3 immunoprecipitated from HEK293T cells expressing a short hairpin RNA against TRAPPC8 (shTRAPPC8) pulls down a similar amount of TRAPPC4 to that from control cells (Fig 6D). These data indicate that TRAPPC8 mediates the interaction between TBC1D14 and the rest of TRAPP.

Intriguingly, TRAPPC8 is the mammalian orthologue of yeast Trs85, the autophagy-specific subunit of TRAPPIII in yeast (Nazarko *et al*, 2005; Barrowman *et al*, 2010). The existence of a TRAPPIII-like complex in mammals has been suggested previously using over-expressed proteins (Behrends *et al*, 2010; Bassik *et al*, 2013), and it may contain TRAPPC8 and the mammal-specific subunit TRAPPC12/TTC15 (Bassik *et al*, 2013). By immunoprecipitating endogenous TRAPPC12, we showed that endogenous TRAPPC8 was co-precipitated (Fig EV2C). Using size exclusion chromatography, we showed that TRAPPC8, TRAPPC12 and the core subunit TRAPPC4 co-elute in a complex of approximately 600 kDa (Fig EV2D). TRAPPC4 is also present in a lower molecular weight complex, which may represent core TRAPP complexes (Fig EV2D). These data provide the first evidence of the existence of a TRAPPIII-like complex in mammalian cells at the endogenous protein level.

### TRAPPC8 is required for Golgi integrity, secretion and autophagy

Given the interaction between TBC1D14 and TRAPPC8, we reasoned that depletion of TRAPPC8 might have the same effect on membrane traffic as overexpression of TBR. We analysed transferrin recycling in TRAPPC8-depleted cells and found no effect compared to Rab11 (Fig 7A) as previously reported (Ishii *et al*, 2013); however, similar to the effect of Rab1A/B siRNA, TRAPPC8 depletion disrupted Golgi morphology in HEK293A cells (Fig 7B) and impaired constitutive secretion in HeLa C1 cells (Fig 7C).

These data confirm that TRAPPC8 is needed for maintenance of Golgi integrity as has been shown for TRAPPC12 and other large TRAPP subunits (Yamasaki *et al*, 2009; Scrivens *et al*, 2011) and demonstrate a role for TRAPPC8 in normal Golgi function.

**Figure 5. TBR expression inhibits autophagy at an early stage.**

A   HEK293A cells expressing GFP, GFP-TBC1D14 224–669 or GFP-TBR were treated in duplicate with EBSS, EBSS plus 100 nM BafA1 or not for 2 h, lysed and subjected to immunoblotting for LC3B, tubulin and GFP. The amount of LC3B/tubulin for each condition from three independent experiments is shown on the bar graph, ± s.e.m. *P < 0.05, **P < 0.01, one-way ANOVA with Sidak's multiple comparison test.

B   HEK293A GFP-LC3B (2GL9) cells were transfected with mCherry or mCherry-TBR, treated with EBSS for 2 h, fixed and analysed by confocal microscopy.

C   HEK293A cells were treated as in (B), fixed, stained for WIPI2 and analysed by confocal microscopy.

D   HEK293A cells stably expressing GFP-DFCP1 were treated as in (B), fixed and analysed by confocal microscopy.

Data information: In (B–C), scale bars, 20 μm. For quantification of (B–D), 10 fields of view containing transfected cells were imaged for each of three experiments, and the number of GFP-LC3B, WIPI2 or GFP-DFCP1 puncta per cell enumerated using Imaris software (Bitplane). Error bars ± s.e.m., *P < 0.05, ***P < 0.001, one-way ANOVA with Sidak's multiple comparison test.

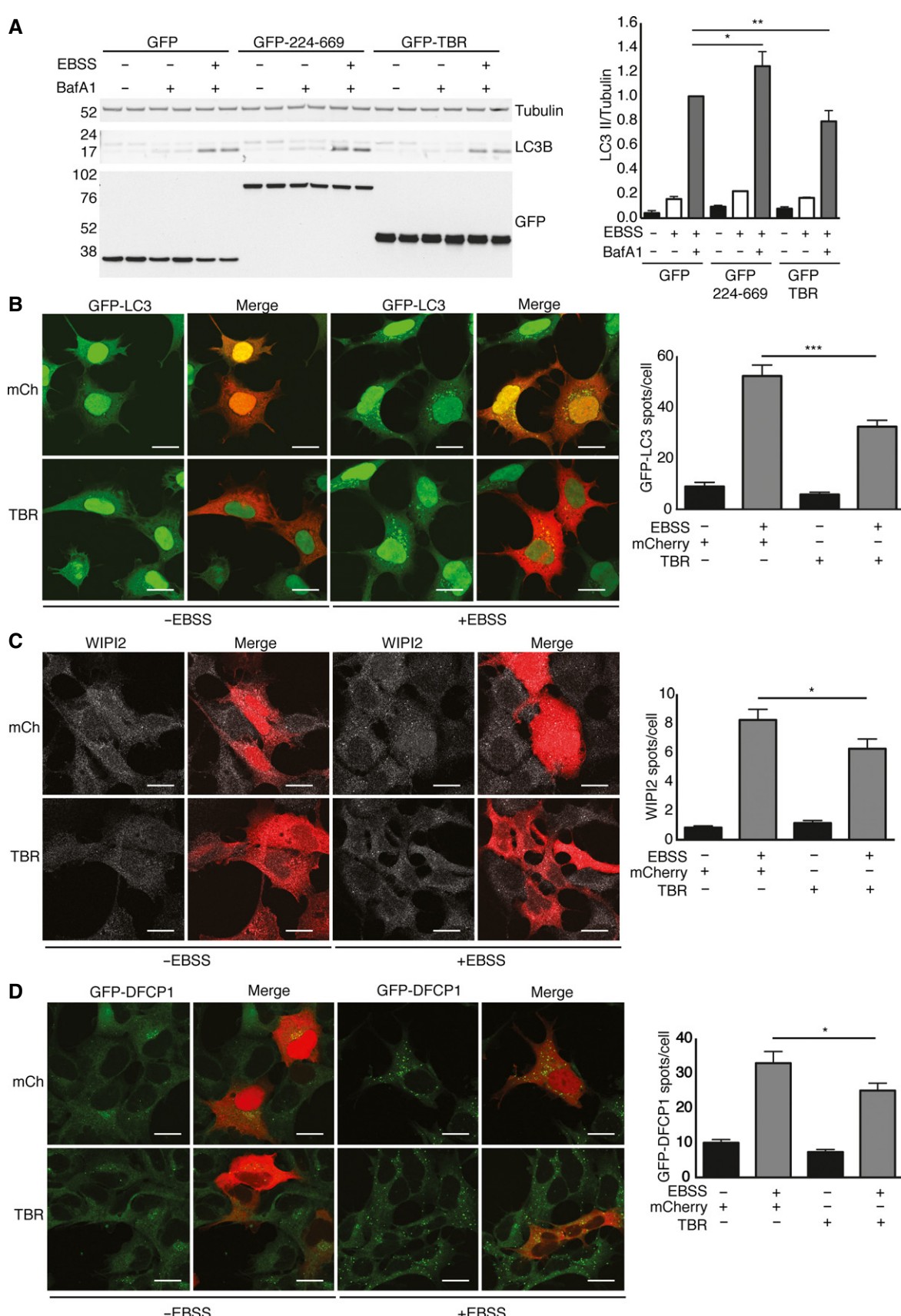

Figure 5.

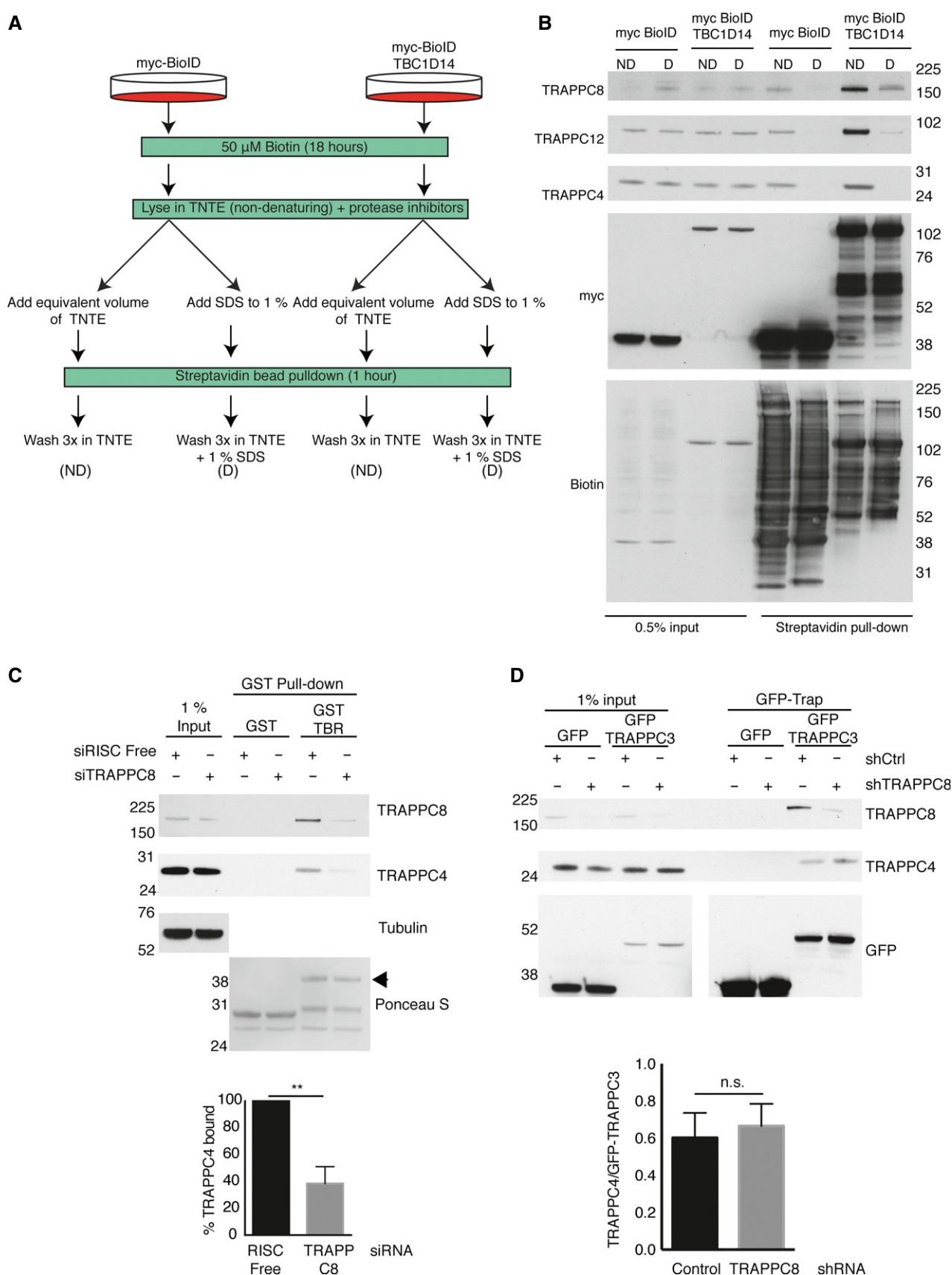

**Figure 6.**

**Figure 6.  TRAPPC8 is required for TBC1D14 to bind the TRAPP complex.**

A  Overview of BioID technique used to determine TRAPP subunit most proximal to TBC1D14. D = denaturing conditions, ND = non-denaturing conditions.
B  Samples prepared according to (A) ($8 \times 10^7$ HEK293A cells per transfection) were lysed in TNTE, split equally and denatured with 1% SDS (D) or not (ND) and subjected to 1-h streptavidin resin pulldown at room temperature. The bound proteins were eluted in 2× Laemmli sample buffer plus 3 mM Biotin and immunoblotted for the indicated proteins.
C  siRNA-transfected control cells (RISC free) or TRAPPC8 knockdown cells (TRAPPC8) were lysed in TNTE and used in pulldown assays with immobilised recombinant GST or GST-TBR. Bound proteins were subjected to immunoblot analysis using TRAPPC8, TRAPPC4 and tubulin antibodies. Ponceau staining was used to verify loading of the GST fusion proteins (GST-TBR indicated with arrowhead). The bar chart expresses the relative amount of TRAPPC4 isolated from siTRAPPC8 compared to siRISC-free control cells from three independent experiments, and error bars are ± s.e.m. **$P < 0.01$, unpaired $t$-test.
D  HEK293T cells stably depleted of TRAPPC8 (shC8) or not (shCtrl) were transfected with constructs encoding GFP or GFP-TRAPPC3 and subjected to GFP-Trap IP. Precipitated proteins were immunoblotted for GFP, TRAPPC4 and TRAPPC8. Bar chart: the amount of TRAPPC4 isolated was normalised to the amount of GFP-TRAPPC3 precipitated from three independent experiments and plotted ± s.e.m., ns = non-significant, unpaired $t$-test.

Importantly, these trafficking phenotypes replicate those seen for the TBC1D14 TBR domain.

As TBC1D14 overexpression fragments the Golgi and inhibits RAB1-dependent processes, we tested whether TBC1D14 overexpression results in a change in RAB1 activation (i.e. GTP binding) by analysing binding of endogenous RAB1 to its effector, Golgin-84 (GOLGA5; Diao *et al*, 2003; Satoh *et al*, 2003). We found that endogenous RAB1B bound to GFP-Golgin-84 to the same extent in cells expressing myc-TBC1D14 compared to control cells, suggesting that TBC1D14 alone cannot promote GTP binding to RAB1B (Fig EV2E). However, TBC1D14 depletion resulted in a significant reduction in RAB1B bound to GFP-Golgin-84, comparable to depletion of the core TRAPP subunits TRAPPC4 and TRAPPC8 (Fig EV2F). These data indicate that TBC1D14 is necessary but not sufficient for normal RAB1-GEF activity. Moreover, these findings confirm that the Golgi fragmentation phenotype seen on TBC1D14 overexpression (Fig 4C) is not due to a reduction of RAB1B-GTP levels and is caused by TRAPP mis-localisation.

As TRAPPC8 mediates the interaction between TBC1D14 and the rest of the TRAPP complex, we decided to investigate the effect of TRAPPC8 depletion on autophagy. TRAPPC8 depletion resulted in a striking reduction in GFP-LC3B and endogenous WIPI2 puncta, and a small but significant reduction in GFP-DFCP1 spot formation (Fig EV3A–C). These phenotypes were similar to that seen for RAB1 depletion, although RAB1 depletion inhibited GFP-DFCP1 spot formation more effectively (Fig EV3C). These data show that the TRAPP complex acts early in autophagosome formation. This finding is reinforced by the known role of the GEF target of TRAPP, RAB1, in autophagy and the formation of DFCP1-positive structures (Zoppino *et al*, 2010; Mochizuki *et al*, 2013).

Although we can detect significant defects in early autophagy markers when TRAPPC8 is depleted (Fig EV3C), and these

correlate with the phenotype seen on expression of TBR (Fig 5), there is no effect on LC3 lipidation after TRAPPC8 depletion, and only a mild effect upon RAB1 depletion (Fig EV4A). To investigate why this discrepancy may occur, we analysed the morphology of the ER-Golgi intermediate compartment (ERGIC) in cells lacking TRAPPC8 or overexpressing TBR (Fig EV4B and C). The ERGIC has recently been shown to be important in LC3 lipidation (Ge *et al*, 2013), and as TRAPPC8 depletion and TBR overexpression both fragment the Golgi, we tested whether the ERGIC was also disrupted. Remarkably, TRAPPC8 depletion only mildly perturbed the juxtanuclear localisation (Fig EV4B), while TBR overexpression severely compromised ERGIC structure with the juxtanuclear ERGIC53 scattered as puncta throughout the cytoplasm (Fig EV4C). These findings may explain the discrepancy between the effects on the formation of LC3-positive puncta and the lipidation of LC3; lack of a functional juxtanuclear ERGIC in TBR-expressing cells may result in a more profound effect on LC3 lipidation, as has been shown in other systems (Ge *et al*, 2013).

## TRAPPC8 function is needed to maintain normal ATG9 traffic

Previous studies in yeast have shown that Trs85/TRAPPIII is needed for the normal progression of autophagy. There are two models proposed—firstly, that Trs85 is present on Atg9-positive membranes, and required to tether them together at the PAS to form the phagophore (Lynch-Day *et al*, 2010; Kakuta *et al*, 2012; Yamamoto *et al*, 2012), and secondly, that TRAPPIII regulates the trafficking itinerary of Atg9, retaining it at the Golgi in the fed state and providing a reservoir of Atg9 for the induction of starvation (Shirahama-Noda *et al*, 2013). Given our data in mammalian cells on the effect of TBC1D14 (Longatti *et al*, 2012) and TRAPP on autophagy and secretion, we

**Figure 7.  Loss of TRAPPC8 disrupts Golgi structure and function.**

A  HEK293A cells transiently transfected with RISC-free control siRNA or siRNA duplexes directed against TRAPPC8 and RAB11A and B (knockdown confirmed by immunoblot analysis) were fed with Alexa-647 transferrin for 15 min in full medium, and the fluorescent transferrin chased out for the indicated time periods. At the end of the time course, cells were trypsinised, fixed and the Alexa-647 fluorescence analysed by flow cytometry. Results are plotted as percentage of transferrin fluorescence at $t = 0$ for the RISC-free control cells, expressed as the mean of three independent experiments ± s.e.m.
B  Cells transiently transfected with RISC-free control siRNA, or siRNA duplexes directed against TRAPPC8 or RAB1A and B were stained for the *cis*-Golgi markers GM130 or RAB1B and analysed by confocal microscopy. Scale bars, 20 µm in main panels, 10 µm in inset panels. Bar graph: > 100 cells per siRNA were scored for fragmented or normal juxtanuclear Golgi stacks from three independent experiments. Error bars ± s.e.m., *$P < 0.05$, **$P < 0.01$, one-way ANOVA with Sidak's multiple comparison test.
C  HeLa C1 cells transiently transfected with RISC-free control siRNA or siRNA duplexes directed against TRAPPC8 or RAB1A and B (knockdown confirmed by immunoblot analysis) were treated with D/D solubiliser for the indicated time periods, trypsinised, fixed and their GFP content analysed by flow cytometry. The line graph shows the GFP fluorescence as a percentage of $t = 0$, ± s.e.m. from three independent experiments.

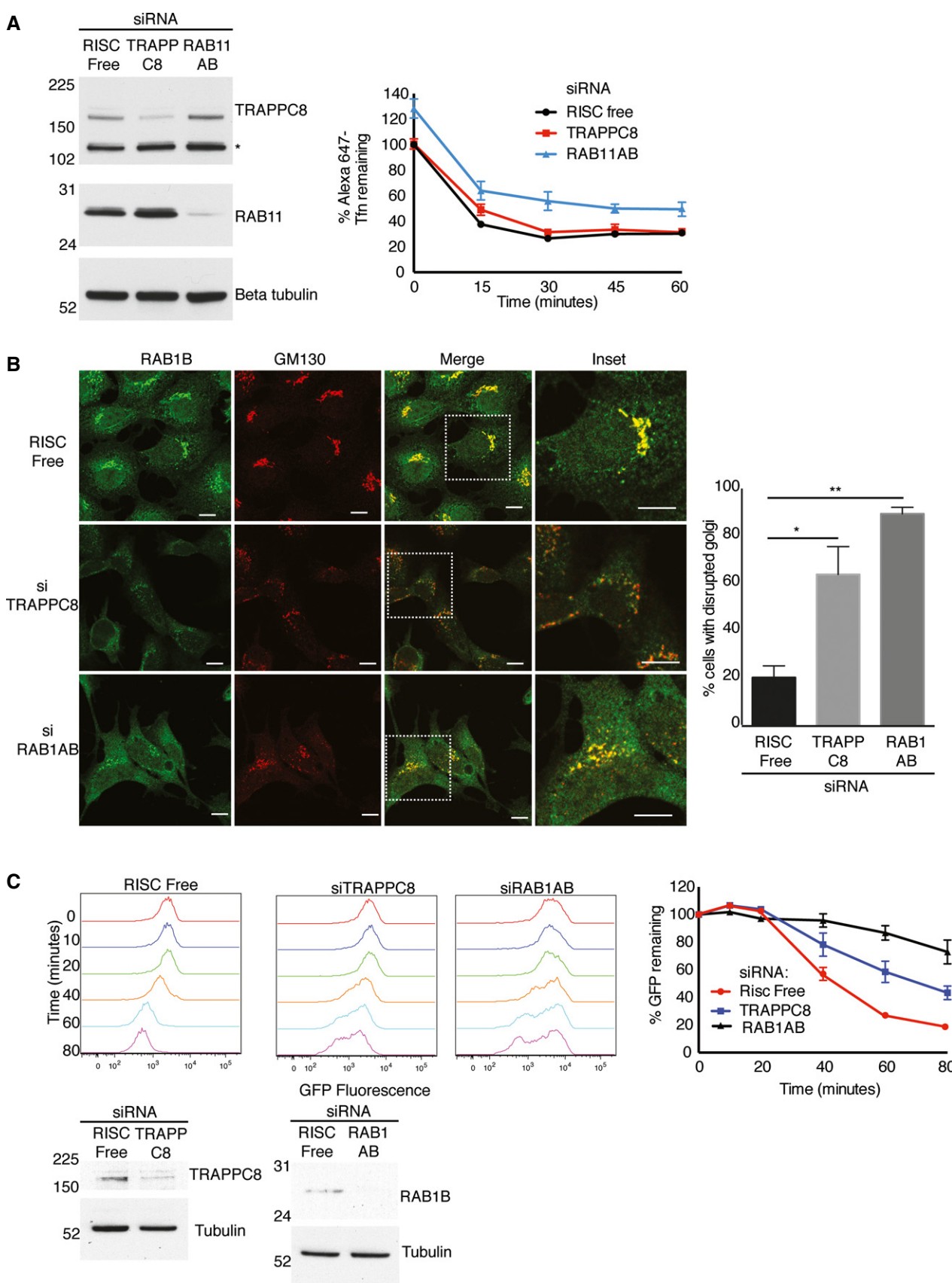

**Figure 7.**

**Figure 8.  TRAPP function is required for normal ATG9 traffic.**

A   ATG9-positive membranes were immunoisolated from approximately $2 \times 10^7$ fed (F) or starved (S) HEK293A cells using a monoclonal hamster anti-ATG9 (ATG9) or control hamster IgM and immunoblotted for ATG9 using a rabbit polyclonal antibody, TRAPPC8, TRAPPC4 and actin.
B   HEK293A cells transfected with GFP or GFP-TBR (green) were stained for ATG9 using hamster anti-ATG9 (red) and analysed by confocal microscopy. Scale bars, 20 μm.
C   HEK293A cells transiently transfected with RISC-free siRNA or siRNA against TRAPPC8 were stained for RAB1B (green) and ATG9 using hamster anti-ATG9 (red) and imaged using confocal microscopy. Scale bars, 20 μm.
D   Stills at $t$ = 30 min from time lapse imaging of 9B9 cells (stably expressing GFP-LC3 and mRFP-ATG9) expressing either CFP or CFP-TBR (blue). Scale bars, 20 μm. Inset panels show mRFP-ATG9 and GFP-LC3 contact events at the indicated time points; arrowheads indicate contact events. Bar chart shows number of times a GFP-LC3 spot was within 1 μm of an mRFP-ATG9 spot (a contact event), expressed as total mRFP-ATG9 contacts per GFP-LC3 spot ± s.e.m., $n$ = 18 cells per condition pooled from 4 independent experiments.

hypothesised that TBR overexpression or TRAPPC8 depletion affected autophagy via alteration of ATG9 localisation or trafficking.

Immunoisolation of ATG9-positive membranes from HEK293A cells revealed that TRAPP subunits including TRAPPC8 were co-isolated with ATG9, regardless of the induction of autophagy (Fig 8A). This is similar to the situation observed in yeast, where Trs85 is enriched in Atg9 vesicles (Kakuta *et al*, 2012) and suggests TRAPP subunits direct ATG9 trafficking. When we disrupted TRAPP function by overexpressing GFP-TBR, we observed that the juxtanuclear ATG9 accumulation observed in HEK293A cells in fed medium was disrupted (Fig 8B). The ATG9 juxtanuclear compartment along with RAB1B-positive structures was also disrupted on TRAPPC8 depletion (Fig 8C), indicating that TRAPP function is required for ATG9 retention at the Golgi. We investigated the effect of TRAPPC8 depletion on a second Golgi-localised protein involved in autophagy, GFP-DFCP1, and found that the juxtanuclear accumulation of GFP-DFCP1 observed in fed cells was diminished after TRAPPC8 depletion, along with a subtle fragmentation of GM130-positive structures (Fig EV5). This shows that TRAPPC8 is required for the trafficking of Golgi-localised autophagy regulators.

We have previously shown that ATG9-positive vesicles transiently contact growing LC3B-positive phagophore and autophagosomes without actually fusing with them, and have speculated that lipid or protein delivery may be one of the functions of this compartment (Orsi *et al*, 2012). To investigate whether TBR expression, and thus TRAPP inhibition, prevents these transient contacts, we expressed CFP-TBR or CFP alone in 9B9 cells, which stably express mRFP-ATG9 and GFP-LC3B (Orsi *et al*, 2012), and analysed the number of times mRFP-ATG9 and LC3-positive structures came into contact (< 1 μm apart) during a 1-h starvation by live confocal microscopy (Movies EV1 and EV2, still images in Fig 8D). We could not observe any differences between the number of contacts in CFP-TBR- or CFP-expressing cells (Fig 8D). These data show that loss of TRAPP activity affects the steady-state distribution of the ATG9 compartment, not its ability to tether with LC3-positive membranes.

**TBR-induced ATG9 dispersion is independent of ULK1**

Previous work from our group has demonstrated that ULK1 kinase is required for the amino acid starvation-induced redistribution of ATG9 to peripheral compartments; on starvation, ULK1-depleted cells retain a juxtanuclear accumulation of ATG9 (Young *et al*, 2006; Orsi *et al*, 2012). As disruption of TRAPP function in fed cells results in dispersion of juxtanuclear ATG9, we wanted to confirm whether this dispersion requires ULK1.

We transfected cells with non-targeting RISC-free siRNA or siRNA against ULK1 and subsequently transfected cells with plasmids

encoding GFP or GFP-TBR. Analysing these cells by confocal microscopy, we found, as anticipated, that ULK1-depleted cells expressing GFP alone did not redistribute ATG9 on starvation (Fig 9A, upper panels). As expected, RISC-free siRNA-treated cells expressing GFP-TBR had severely reduced juxtanuclear ATG9 staining in fed conditions (see Figs 8B and 9A, middle panel). Remarkably, this was maintained in ULK1-depleted cells and upon starvation. These data suggest that TBC1D14 and TRAPP may act upstream of ULK1 in regulating ATG9 traffic, perhaps by maintaining normal traffic into the Golgi. As TRAPP is a RAB1 GEF (Sacher *et al*, 1998; Barrowman *et al*, 2010), we decided to test whether loss of RAB1 activity phenocopied the TBR overexpression effect on ATG9 traffic. Indeed, overexpression of myc-RAB1B S22N, a GDP-locked dominant negative form of the protein (Alvarez *et al*, 2003), fragmented the juxtanuclear ATG9 compartment, and ULK1 depletion did not prevent this dispersion (Fig 9B, middle and lower panels). Conversely, expression of wild-type myc-RAB1B did not result in dispersion of ATG9 in ULK1-depleted cells, in fed or starved conditions (Fig 9B, upper panels). These data support the notion that TRAPP- and RAB1-dependent effects on ATG9 trafficking are upstream of ULK1 activity, and form part of a more fundamental trafficking process.

## Discussion

Our previous analysis of putative RabGAP proteins affecting LC3 lipidation identified TBC1D14 as a negative regulator of autophagy (Longatti *et al*, 2012). We show here that the mammalian version of the TRAPPIII complex binds to TBC1D14, and that this interaction is mediated by an N-terminal 103 amino acid stretch in TBC1D14, which we call the TBR, and TRAPPC8, the mammalian orthologue of yeast Trs85 (Scrivens *et al*, 2011).

Importantly, we show endogenous TRAPPC8 is a subunit of a "TRAPPIII-like" complex in mammalian cells and behaves as a positive regulator of autophagy, with its depletion reducing the number of autophagic puncta. Given the interaction between TRAPP and TBC1D14 and building on our previous finding of TBC1D14's negative regulation of autophagy (Longatti *et al*, 2012), we show that recruitment of TRAPPIII by TBC1D14 tubulates RE and inhibits autophagy.

We have previously demonstrated that TBC1D14 resides on both peripheral RE and Golgi membranes (Longatti *et al*, 2012); however, the tubules generated on TBC1D14 overexpression are positive for RAB1, RAB11 and transferrin (Figs 1 and 2). In support of an interaction of TBC1D14 and TRAPPIII, it has been shown that exchange between the transferrin/RAB11-positive RE and the RAB1-positive ERGIC/early Golgi can occur (Marie *et al*, 2009). The presence of

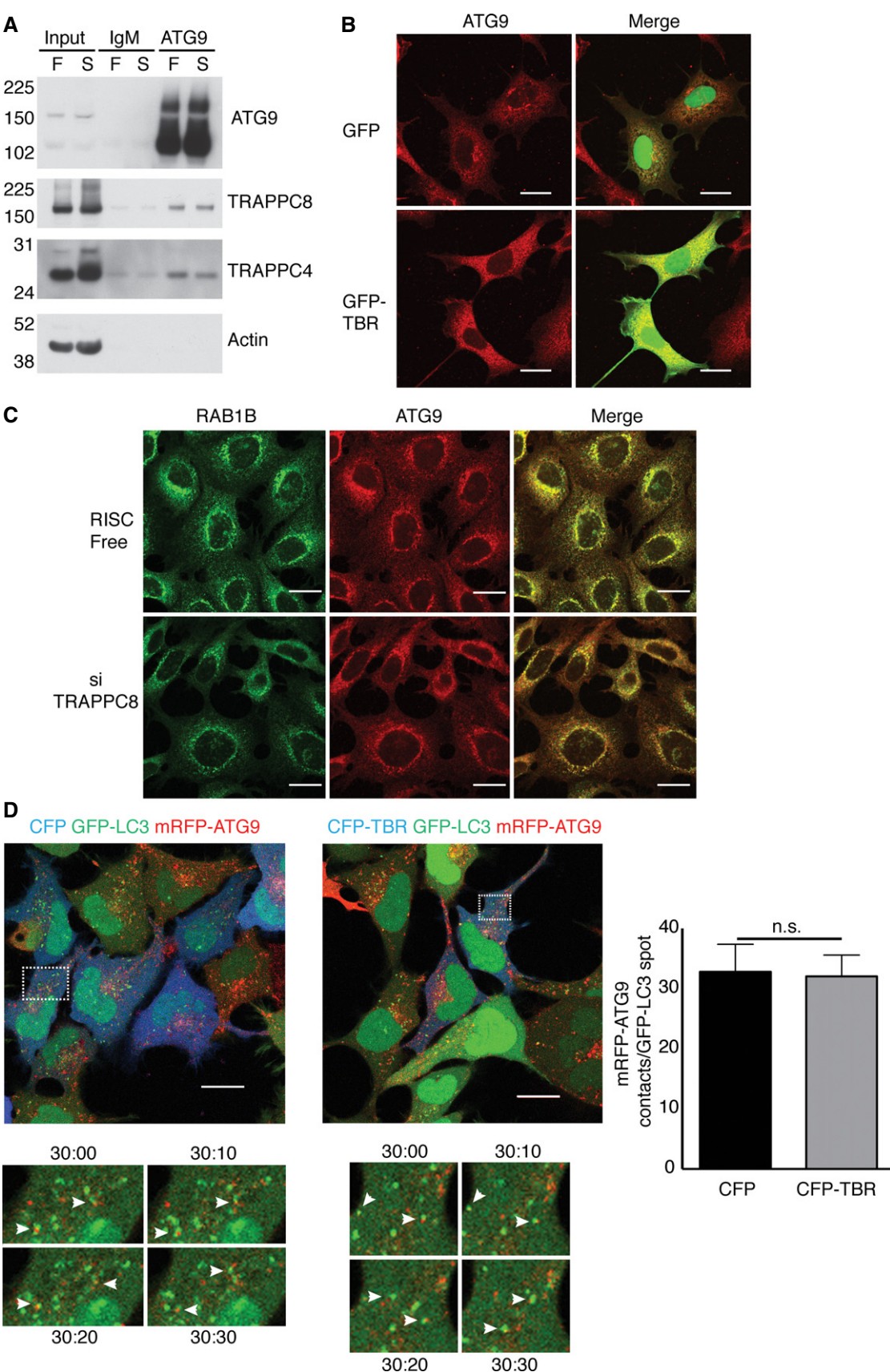

**Figure 8.**

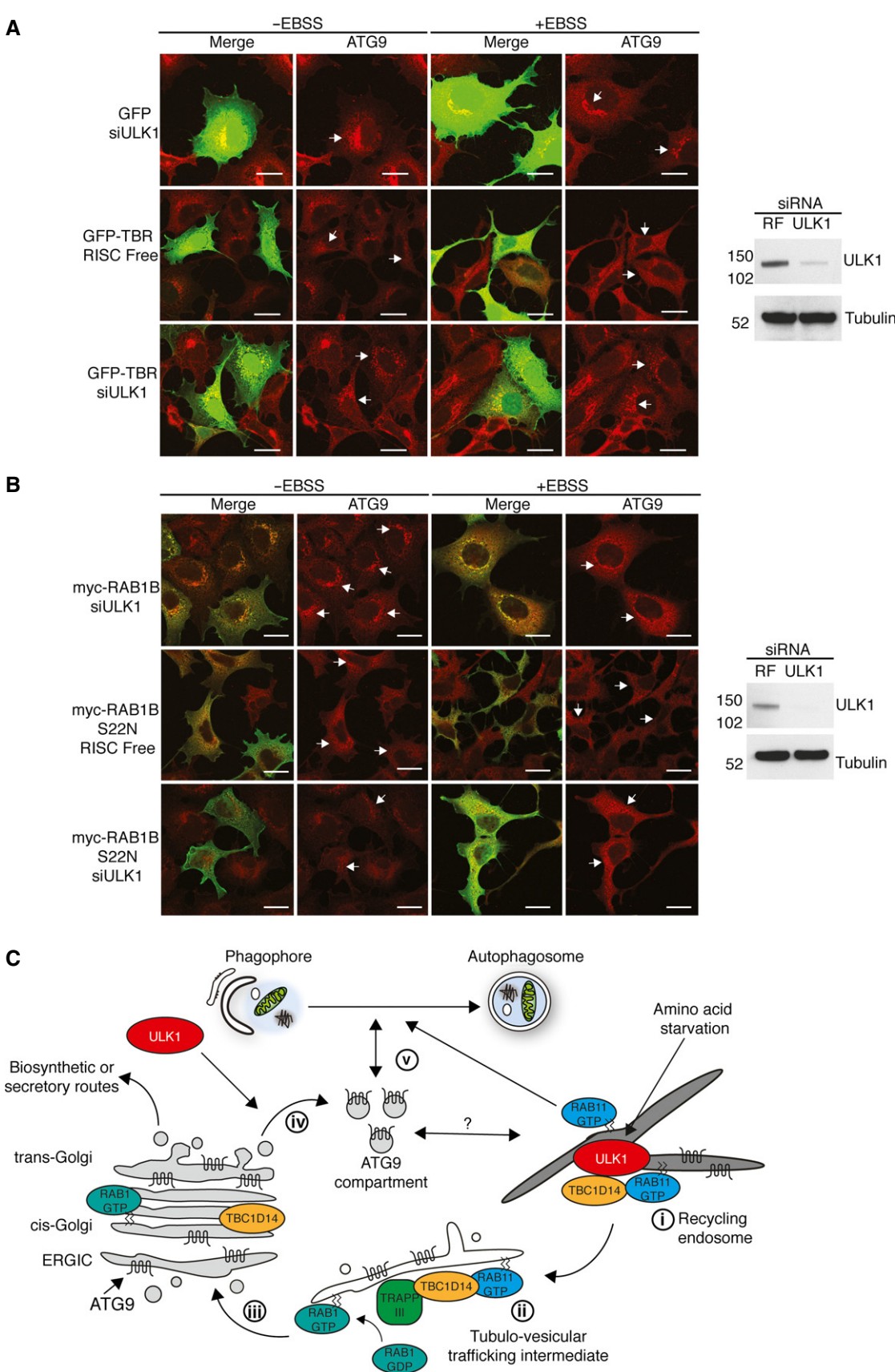

**Figure 9.**

**Figure 9.  A TBC1D14- and RAB1-dependent trafficking step is essential for normal ATG9 traffic.**

A   Cells depleted of ULK1 (siULK1) or not (RISC free) were transfected with GFP or GFP-TBR (green), starved (EBSS) or not (−EBSS) and stained for ATG9 using hamster anti-ATG9 (red). Arrows indicate transfected cells. Western blot indicates ULK1 knockdown. Scale bars, 20 μm.

B   Cells depleted of ULK1 (siULK1) or not (RISC free) were transfected with myc-RAB1B or myc-RAB1B (S22N) (green), starved (EBSS) or not (−EBSS) and stained for ATG9 using hamster anti-ATG9 (red). Arrows indicate transfected cells. Western blot indicates ULK1 knockdown. Scale bars, 20 μm.

C   Model for the role of TBC1D14 and the TRAPP complex in mammalian autophagy and ATG9 traffic. Recycling endosomes harbour a population of TBC1D14 bound to RAB11, and a population of ATG9 molecules which may traffic to and from the ATG9 compartment. (i) Upon amino acid starvation, TBC1D14 and RAB11 induce a vesicle trafficking step from RAB11- to RAB1-positive membranes at a tubulo-vesicular transport intermediate (ii). This results in recycling of ATG9 to RAB1-regulated Golgi compartments (iii) from where ATG9 can be trafficked to the ATG9 compartment in an ULK1-dependent manner (iv). This maintains the ATG9 compartment under starvation, which can then both contribute to, and promote autophagosome formation from other membrane sources, including the RAB11-positive recycling endosome (v).

endogenous TBC1D14 on both these compartments indicates it plays a key role in exchange between RAB11- and RAB1-positive membranes, via TRAPPIII.

TBC1D14 behaves as a RAB11 effector, but we were previously unable to characterise any enzymatic activity for TBC1D14 (Longatti *et al*, 2012). We show here TBC1D14 acts as a negative regulator of autophagy by modulating the contribution of both RE and Golgi membranes to the growing autophagosome through its interaction with Rab11 and TRAPPIII. TBC1D14 is resident both on peripheral transferrin-positive structures and the Golgi complex in basal conditions. Upon starvation and autophagy activation, TBC1D14 levels increase in the Golgi while ATG9 disperses from the Golgi to the ATG9 compartment, in an ULK1-dependent manner (Young *et al*, 2006; Longatti *et al*, 2012). Our previous finding that TBC1D14 overexpression induced tubules do not harbour ATG9 (Longatti *et al*, 2012) suggests that normally the TBC1D14–TRAPP interaction may be important in ATG9 exit from the ATG9 compartment, and potentially the RE.

Our new findings shed light on the potential function of TBC1D14 and support the model shown in Fig 9C. TBC1D14 at the RE is bound to RAB11. RAB11 bound TBC1D14 interacts with the TRAPPIII complex on a vesicular intermediate, where it activates RAB1. This results in conversion of RAB11-positive membranes to RAB1-positive membranes facilitating endosome to Golgi traffic. This trafficking step is required for a rapid cycling of ATG9 under starvation and maintenance of a ready supply of ATG9 vesicles for further autophagosome formation. TBC1D14 may act to balance direct, bulk contribution of RE to autophagosomes through its RAB11-binding ability, while promoting constitutive ATG9 cycling from the Atg9 compartment through its recruitment of TRAPP and activation of RAB1.

This model has some similarities with that recently proposed for Atg9 traffic in yeast (Shirahama-Noda *et al*, 2013), as TRAPP-dependent traffic is needed to maintain a cycling "store" of Atg9 vesicles to support autophagosome formation. This study also identified a role for the GARP (Golgi associated retrograde protein) complex in returning Atg9 to the Golgi from late endocytic compartments; however, the only identified role for GARP in mammalian autophagy thus far is in autophagosome maturation (Perez-Victoria *et al*, 2010). Despite the lack of an obvious TBC1D14 orthologue, the yeast orthologues of RAB11 (Ypt31/32) have been implicated in TRAPP-dependent trafficking and autophagy, suggesting that elements of this pathway are conserved in yeast (Zou *et al*, 2013).

### A role for mammalian TRAPPIII in autophagy

Our findings are the first to clearly characterise a role for mammalian TRAPP, in particular TRAPPC8, in bulk macroautophagy, and a previous network study supports our conclusions (Behrends *et al*, 2010). As anticipated from findings in yeast, where Trs85 appears to be required to nucleate membranes at the PAS to form the phagophore (Lynch-Day *et al*, 2010; Kakuta *et al*, 2012), mammalian TRAPPIII appears to act at an early stage in autophagy, as disruption of TRAPP function results in reduction in DFCP1-positive omegasomes, which are thought to be functionally equivalent to the single PAS found in yeast cells (Axe *et al*, 2008; Hayashi-Nishino *et al*, 2009; Yla-Anttila *et al*, 2009).

Our findings also clarify the relationship between mammalian TRAPP and ATG9 vesicles. While our data support the notion that the TRAPP/RAB1 system is needed to allow ATG9 to cycle through the endomembrane system, as ATG9 is dispersed under basal conditions upon TRAPP disruption, we also find that isolated ATG9-positive membranes harbour TRAPP subunits. We propose that a TRAPPIII-like complex transiently associates with ATG9-positive membranes, facilitating a switch between RAB11 and RAB1 at a vesicular transport intermediate, allowing ATG9 to cycle through the endomembrane system to return to the Golgi. Our data suggest that a RAB cascade (Rink *et al*, 2005) operates during ATG9 trafficking. However, as TBC1D14 appears to act as an effector for RAB11 rather than a GAP (Longatti *et al*, 2012), how RAB11 is inactivated in this cascade remains to be determined.

### The ER exit machinery is crucial for autophagosome formation

At the molecular level, it is already established that RAB1, one of the key determinants of ER-Golgi and intra-Golgi traffic and also the canonical GEF target of the TRAPP complex (Barrowman *et al*, 2010), is required for autophagosome formation, and in particular omegasome formation (Winslow *et al*, 2010; Zoppino *et al*, 2010; Huang *et al*, 2011; Mochizuki *et al*, 2013), and in our studies, we have shown that RAB1 is in turn necessary for WIPI2 and LC3 puncta formation. Indeed in yeast, the formation of the PAS from Atg9 vesicles depends on the yeast orthologue of Rab1, Ypt1 (Lynch-Day *et al*, 2010; Kakuta *et al*, 2012; Lipatova *et al*, 2012), and there is evidence that Ypt1 can recruit Atg1 to the PAS (Wang *et al*, 2013). Our data support the notion that the ER-Golgi trafficking machinery plays a second conserved role in autophagosome formation, not only creating platforms for and contributing membrane to forming autophagosomes but also by promoting trafficking of ATG9 from compartments, potentially including the RE, to RAB1-positive transport vesicles, thus supporting autophagosome biogenesis (Fig 9C). Further analysis of cells lacking key components of the ER-Golgi trafficking machinery using imaging techniques such as those employed in recent studies (Karanasios *et al*, 2013; Koyama-Honda *et al*, 2013) will help to clarify these findings.

More broadly, our data add to a growing body of evidence that proteins and compartments thought to be required for exit of proteins from the ER and into the Golgi are in fact also regulating autophagy. Both ultrastructural studies (Biazik *et al*, 2015) and biochemical studies have suggested a role for ER exit sites in autophagosome formation, whereby the ERGIC as a source of LC3 lipidation activity contributes to autophagosome formation, (Ge *et al*, 2013). Another recent yeast study has linked ER exit sites to autophagy (Graef *et al*, 2013), and indeed, COPII vesicles seem to be needed for autophagy to progress (Tan *et al*, 2013). In the wider context of membrane trafficking, our study highlights the potential for membrane exchange between RAB11-positive RE and RAB1 regulated early Golgi compartments such as the ERGIC (Marie *et al*, 2009) in regulating autophagosome formation. The identification of other cargos that traverse the RAB11-RAB1 route will increase our understanding of how this trafficking step contributes to autophagy.

# Materials and Methods

### Cell culture

Cell lines are described in detail in the Appendix Supplementary Methods and were maintained in Dulbecco's modified Eagle's medium (DMEM, Sigma-Aldrich) supplemented with 10% foetal bovine serum (FBS) and 4 mM glutamine at 37°C and 10% $CO_2$. Amino acid starvation was induced by washing cells three times in Earle's balanced saline solution (EBSS) and incubating in EBSS, or EBSS + 100 nM bafilomycin A1 (BafA1; Calbiochem) to block lysosomal acidification, for 2 h. For recombinant protein expression for Western blot and immunoprecipitation, cells were transfected using Lipofectamine 2000 (Life technologies) according to manufacturer's instructions. For recombinant protein expression for confocal analysis, cells were transfected with Fugene HD (Promega) according to manufacturer's instructions. For siRNA transfection, cells were transfected using a forward transfection strategy using two siRNA hits (Day 1—Oligofectamine (Life technologies) and Day 2—Lipofectamine 2000), except for ULK1 depletion where reverse transfection using Lipofectamine 2000 was used.

### Constructs

GFP-TRAPPC3 was a gift from A. DiMatteis and R. Venditti (TIGEM, Naples, Italy). Myc-RAB1B wt and myc-RAB1B S22N were kind gifts from Cecilia Alvarez (University of Cordoba, Spain). Myc-BioID was from Addgene and was originally generated by the Roux laboratory (Roux *et al*, 2012). pECFP-C1 (Clontech) was a gift from L. Ombrato (Francis Crick Institute). GFP-Golgin 84 was gift from Martin Lowe (University of Manchester, UK). All other constructs were previously described (Longatti *et al*, 2012) or generated during this study.

### Confocal microscopy

Cells grown on coverslips were fixed in 3% PFA, or 2% PFA, 75 mM Lysine-HCl, 10 mM $NaIO_4$ in 375 mM sodium phosphate buffer pH 6.2 (Marie *et al*, 2009), quenched with 50 μM $NH_4Cl$ and permeabilised in 50 μg/ml digitonin or 0.2% Triton X-100 in PBS. Cells were blocked in 5% BSA in PBS and incubated with appropriate primary and secondary antibodies in 5% BSA for 1 h each in a humidified chamber. Fixed cells were analysed using a Zeiss Upright 710 confocal microscope with a 63× objective lens. For spot counting, a 40× objective was used and images were analysed using Imaris software (Bitplane). For quantification, cells were located using Hoechst DNA dye and subsequently imaged in other channels.

### Live cell imaging

9B9 cells (Orsi *et al*, 2012) grown in 35-mm glass-bottomed dishes (Mattek corporation) were starved in EBSS + 30 mM HEPES for 15 min and filmed for a further 60 min at 37°C using a Zeiss Inverted 880 microscope and 63× objective lens with a 0.6 μm Z-section, and images acquired every 10 s. Proximity of GFP-LC3 spots to mRFP-ATG9 spots was quantified using Imaris software (Bitplane) with the Distance Transformation Matlab plugin, with GFP-LC3 spots < 1 μm from an mRFP-ATG9 spot being scored as a contact event.

### GST pulldown

Approximately $8 \times 10^6$ cells were washed with twice with PBS, harvested and lysed in 800 μl TNTE plus complete protease inhibitor cocktail (PI) (Roche). The lysates were divided between GST and GST-TBR resin and incubated for 2 h with rotation at 4°C. Bound proteins were washed three times in TNTE plus PI and eluted from the beads with 30 μl 2× Laemmli sample buffer.

### Isolation of Atg9-positive membranes

HEK-293 cells were treated with full medium (DMEM) or EBSS for 2 h. Cells were then washed in PBS and harvested by centrifugation. Pellets were suspended in an isotonic buffer and passed through a 25G needle before pre-clearing by centrifugation. Supernatants were used for incubation overnight at 4°C with non-specific IgM or Hamster anti-Atg9 antibody-coupled beads. The ATG9-positive membranes on the beads were then washed and eluted by peptide competition. All isolated ATG9-positive membranes were resuspended in Laemmli sample buffer for gel electrophoresis and immunoblotting.

### Flow cytometry analysis of transferrin recycling and bulk secretion

For analysis of transferrin recycling, HEK293A cells were grown in 6-well cluster plates. The cells were fed fluorescent transferrin (Alexa-647-Tfn, Life technologies) for 15 min in DMEM, then the Alexa-647-Tfn chased out of the cells with fresh DMEM for the indicated time periods. For analysis of bulk secretion, HeLa C1 cells (Gordon *et al*, 2010) were grown in 6-well cluster plates. Cells were treated with D/D solubiliser (Clontech), or D/D solubiliser plus 10 μg/ml Brefeldin A (BFA, Sigma-Aldrich) for the indicated time periods. On completion of the time course, the cells were trypsinised, the trypsin inactivated with DMEM, the cells were fixed in 4% PFA, washed into PBS and fluorescence analysed using an LSRII or Fortessa flow cytometer (Beckton Dickinson), gating for single cells with forward and side scatter, and fluorescent tags on overexpressed proteins (GFP or CFP) where necessary.

**Western blot quantification**

Western blots were quantified using ImageJ software (NIH).

**Data analysis**

All graphs were plotted and statistical tests performed using Graphpad Prism software.

**Expanded View** for this article is available online.

## Acknowledgements

The authors thank A. Peden, C. Alvarez, M. Lowe, L. Ombrato, K. Roux, R. Venditti and A. De Matteis for cell lines and constructs. The authors acknowledge the assistance of the Francis Crick institute Flow Cytometry, Light Microscopy and Protein Purification facilities with several experiments. The authors thank F. Barr for helpful discussion and the laboratory members for useful input throughout the project. This work was supported by the Francis Crick Institute, which receives its core funding from Cancer Research UK, the UK Medical Research Council and the Wellcome Trust (CAL, DJ, DF, APS, SAT), and grants from the German Research Foundation (BE 4685/1-1) (SN, CB).

## Author contributions

CAL designed and performed most of the experiments and wrote the manuscript. SN and CB generated the stable shRNA cells and provided reagents and advice. DF and APS performed the mass spectrometry analysis of the BioID experiments. DJ performed the ATG9 immunoisolation experiments. SAT directed the research, performed the GST pulldown in Fig 1 and wrote the manuscript.

## Conflict of interest

The authors declare that they have no conflict of interest.

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
