## [Review Process File · The EMBO Journal]

Manuscript EMBO-2015-92695

TBC1D14 regulates autophagy via the TRAPP complex and ATG9 traffic

Christopher Lamb, Ms. Stefanie Nühlen, Delphine Judith, David Frith, Ambrosius Snijders, Christian Behrends and Sharon Tooze

Corresponding author: Sharon Tooze, The Francis Crick Institute -Member -EBM

Review timeline:

Submission date:	30 July 2015
Editorial Decision:	31 August 2015
Revision received:	12 November 2015
Editorial Decision:	24 November 2015
Accepted:	26 November 2015

Editor: Andrea Leibfried

Transaction Report:

1st Editorial Decision

31 August 2015

Thank you for submitting your manuscript entitled 'TBC1D14 regulates autophagy via the TRAPP complex and ATG9 traffic'. I have now received the reports from all referees, which are copied below.

As you can see, the referees appreciate your findings and referee #1 clearly supports publication pending satisfactory revision. Referee #2 notes that the proposed mechanism needs to be supported by additional data and information. Most importantly, both referees state that your previous findings on TBC1D14 need to be better incorporated in your model, and that the model needs further clarification.

Given the very constructive comments provided, I would like to offer you to submit a revised version of the manuscript, addressing all concerns of the referees. All points mentioned by the referees seem to be straightforward to address, but please contact me in case of questions regarding the revision of your manuscript.

Thank you for the opportunity to consider your work for publication. I look forward to your revision.

REFEREE REPORTS

Referee #1:

This is a significant and conclusive study, and the presented data support author's conclusions. The reasons for why a study is seen as an important advance is as follows: The authors identify a connection between a Rab GAP (TBC1D14), previously linked to autophagy as its regulator via Rab11, and TRAPP (a GEF for Rab1). TBC1D14 molecularly links early secretory pathway and recycling endosome with autophagosome biogenesis. Whereas TRAPP and Rab11 have been studied before, the link between the two represents an important finding and provides significant advance in functional links between different compartments, previously implicated as isolated entities in autophagosome formation. The role of TBC1D14 here is a key linchpin (albeit undersold by the authors) in understanding how these different pathways are coordinated, and represents a highly pertinent and novel information. The experimental results are simply gorgeous in execution and presentation.

Some points to address:

1. It may be a vast oversimplification to name only p62 and NDP52 as autophagic receptors.
2. Fig. 3. Some inputs are low (e.g. GFP-224-669). Can authors address this?
3. The use of GFP-FM4-hGH and rapamycin - while elegant in terms of block release, is the rapamycin DD/ solubilizer completely devoid of action in autophagy induction via Tor etc? Can the authors rule this out?
4. Can the analysis of the effects of TRAPPC8 on secretion be a bit more defined, relative to effects on autophagy?
5. Can antibodies used for endogenous Atg9 be described a bit better? There were hamster and rabbit antibodies, against Atg9, and it is not always clear in immune-based analyses (biochem, IF) which antibody and under what conditions was it used. Also, have these been compared to commercial antibodies?
6. The model is nice. However, it seems to shift completely the role of TBC1D14 from Rab11 to TRAPP. A missed opportunity to connect the two compartments here in a functional and sequential way in the context of autophagy? The way the manuscript reads, it gives an impression that the authors think that their previously published work with Rab11 and TBC1D14 is not as relevant. Is this correct? And yet, this reviewer believes that this connection is a key value of this study, as stated in the overall assessment.
7. This is not important, more of an en passant remark, plus I don't know how to say this sounding serious about it. Try as I may, I could not say "TBC1D14" without tripping over my tongue. Is there a way to give a more functional (and easier) name to this protein now that it has been linked to autophagy in multiple ways?

Referee #2:

The group of Sharon Tooze has previously identified TBC1D14 as a RAB11-binding protein that acts as negative regulator of autophagy. Lamb and co-workers now report that TBC1D14 interacts with the TRAPP complex through interaction of an N-terminal region (residues 120-223) with the TRAPPC8 subunit. They also report that TRAPPC8 is required for Golgi integrity, secretion, autophagy, and juxtannuclear localization of ATG9 (in fed cells, independently of ULK1), and that overexpression of the TRAPP-binding region of TBCD14 phenocopies TRAPPC8 downregulation. A model is proposed whereby TBC1D14 and TRAPP^{III}, via activation of RAB1, regulate a constitutive trafficking step from peripheral recycling endosomes to the early Golgi to maintain the cycling pool of ATG9 required for autophagy.

Overall the manuscript contains solid data that are backed up by adequate quantifications. The most novel contribution is the identification of an interaction between TBC1D14 and TRAPP and the evidence that this interaction is important for autophagy regulation. However, the mechanism for this regulation remains incompletely characterized.

Specific remarks:

1. A central conclusion from this study is that TBC1D14 serves to recruit TRAPPIII to the ERGIC to activate RAB1. However, this needs to be verified by studying RAB1 activation under conditions of TBC1D14 depletion and overexpression.
2. The model in Fig. 9c proposes that Rab11 not only localizes to RE but also to ERGIC and cis-Golgi elements. However, the evidence is not convincing on this point. In Fig. 2A it is possible to observe a few structures that apparently contain both RAB11A and RAB1B in cells transfected with GFP-TBC1D14, but if such a co-localization can only be observed under GFP-TBC1D14 overexpression conditions its biological relevance remains uncertain. In addition, the resolution of the light microscope is insufficient to resolve distinct but juxtaposed vesicles, and immunoelectron microscopy would be required to conclude whether RAB11A and RAB1B truly co-localize on ERGIC and Golgi membranes.
3. The model in Fig. 9c implies that TRAPP III is a positive regulator of autophagy because it activates RAB1, a GTPase required for autophagy. This is confusing in light of previously published results from the Tooze lab that place TBC1D14 as an inhibitor of autophagy.

1st Revision - authors' response

12 November 2015

Referee #1:

This is a significant and conclusive study, and the presented data support author's conclusions. The reasons for why a study is seen as an important advance is as follows: The authors identify a connection between a Rab GAP (TBC1D14), previously linked to autophagy as its regulator via Rab11, and TRAPP (a GEF for Rab1). TBC1D14 molecularly links early secretory pathway and recycling endosome with autophagosome biogenesis. Whereas TRAPP and Rab11 have been studied before, the link between the two represents an important finding and provides significant advance in functional links between different compartments, previously implicated as isolated entities in autophagosome formation. The role of TBC1D14 here is a key linchpin (albeit undersold by the authors) in understanding how these different pathways are coordinated, and represents a highly pertinent and novel information. The experimental results are simply gorgeous in execution and presentation.

We thank the reviewer for their positive assessment of the manuscript and its significance.

Some points to address:

1. It may be a vast oversimplification to name only p62 and NDP52 as autophagic receptors.

We agree, and have changed this on page 3 to read "...autophagy receptors, of which p62 is prototypical" to indicate that many other proteins are known to serve this purpose.

2. Fig. 3. Some inputs are low (e.g. GFP-224-669). Can authors address this?

We repeated the experiment (now part of the revised figure 3B) and increased the amount of GFP-224-669 construct transfected. This resulted in an increased expression of GFP-224-669 to comparable levels to full length GFP-TBC1D14. Of note, GFP-224-669 does not bind to TRAPP subunits under these conditions, but GFP-TBC1D14 and GFP-1-330 (both containing the mapped TRAPP binding region of TBC1D14) bind to TRAPP subunits despite being expressed at a similar level to GFP-224-669.

3. The use of GFP-FM4-hGH and rapamycin - while elegant in terms of block release, is the rapamycin DD/ solubilizer completely devoid of action in autophagy induction via Tor etc? Can the authors rule this out?

We have added two sets of experiments to the new Appendix Figure S1 to address this question. Firstly, we treated HeLa C1 cells with D/D solubiliser, rapamycin or Torin 1 for 80 minutes, which is comparable to the time-course used in our secretion experiments. Immunoblotting for GFP revealed that both D/D solubiliser and Rapamycin induced secretion of GFP-FM4-hGH whereas Torin treatment did not. However, whereas both rapamycin and Torin treatment (as expected) resulted in inactivation of mTOR, as evidenced by phosphorylation of S6 ribosomal protein at mTOR sensitive sites (serines 240/244), the D/D solubilizer did not affect phosphorylation of S6. (figure S1A). This indicates that D/D solubiliser does not inactivate mTOR. Secondly we applied the same treatments to HEK293A cells to investigate whether LC3 I lipidation to LC3 II is induced on D/D solubiliser treatment and found that, unlike both rapamycin and Torin treatment, D/D solubiliser does not induce LC3 lipidation (Figure S1B). These data collectively indicate that D/D solubiliser does not induce autophagy.

4. Can the analysis of the effects of TRAPPC8 on secretion be a bit more defined, relative to effects on autophagy?

We extended our analysis of TRAPPC8 on secretion relative to autophagy by analyzing the localisation of GFP-DFCPI, a second Golgi localised protein involved in autophagosome formation (Axe et al 2008, JCB). We compared the effect of TRAPPC8 knockdown on GFP-DFCPI relative to the Golgi marker GM130 (figure EV5). Our new data show that the juxtannuclear GFP-DFCPI pool is disrupted on TRAPPC8 depletion, along with GM130. This suggests that the regulation of Golgi function and morphology by TRAPPC8 impacts on both Golgi proteins involved in secretion as well as Golgi localised autophagy proteins.

5. Can antibodies used for endogenous Atg9 be described a bit better? There were hamster and rabbit antibodies, against Atg9, and it is not always clear in immune-based analyses (biochem, IF) which antibody and under what conditions was it used. Also, have these been compared to commercial antibodies?

We have clarified the ATG9 antibodies used in Figures 8 and 9 in the corresponding figure legends. We have also added extra information to the supplementary materials and methods sections. The polyclonal rabbit antibody is made in our laboratory and was characterized and referenced in the materials and methods section (Young et al 2006, Journal of Cell Science). The Armenian hamster monoclonal antibody 14F2 was also generated in our group (see Webber and Tooze 2010) and is also commercially available through Abcam and Thermo-Fisher.

6. The model is nice. However, it seems to shift completely the role of TBC1D14 from Rab11 to TRAPP. A missed opportunity to connect the two compartments here in a functional and sequential way in the context of autophagy? The way the manuscript reads, it gives an impression that the authors think that their previously published work with Rab11 and TBC1D14 is not as relevant. Is this correct? And yet, this reviewer believes that this connection is a key value of this study, as stated in the overall assessment.

We thank the referee for their appreciation of our previous data. We agree we did should provide more of a connection, as we agree this is one of the major conclusions of our study. In light of our new data (provided in response to comments by Reviewer 2), which looks at RAB1 activation, and colocalisation of RAB1 and RAB11 after Brefeldin A treatment, we have clarified and modified our model. We propose that TBC1D14 may act to balance direct contribution of membrane to the autophagosome from the peripheral recycling endosome (via its RAB11 binding function) with ATG9 cycling (via its role in TRAPPIII function), and have included this discussion along with the improved model in Figure 9C. Thus, TBC1D14 acts as a negative regulator upon overexpression, or as an inhibitory effector of Rab11 in situations where active Rab1 and TRAPPIII may not yet be engaged. Whether, and how, this balance is regulated remains to be characterized, although the interplay of TBC1D14 with other recycling endosome localised autophagy regulators such as SNX18 (Knaevelsrud et al 2013, JCB) may be important in this regard.

7. This is not important, more of an en passant remark, plus I don't know how to say this sounding serious about it. Try as I may, I could not say "TBC1D14" without tripping over my tongue. Is there a way to give a more functional (and easier) name to this protein now that it has been linked to autophagy in multiple ways?

We appreciate the reviewer's comment. However TBC1D14 has been published about on several occasions using this acronym including a crystal structure for the putative TBC domain (Tempel et al 2008, Proteins), and we feel that changing the name of the protein at this stage would confuse the more general reader.

Referee #2:

The group of Sharon Tooze has previously identified TBC1D14 as a RAB11-binding protein that acts as negative regulator of autophagy. Lamb and co-workers now report that TBC1D14 interacts with the TRAPP complex through interaction of an N-terminal region (residues 120-223) with the TRAPPC8 subunit. They also report that TRAPPC8 is required for Golgi integrity, secretion, autophagy, and juxtannuclear localization of ATG9 (in fed cells, independently of ULK1), and that overexpression of the TRAPP-binding region of TBC1D14 phenocopies TRAPPC8 downregulation. A model is proposed whereby TBC1D14 and TRAPPIII, via activation of RAB1, regulate a constitutive trafficking step from peripheral recycling endosomes to the early Golgi to maintain the cycling pool of ATG9 required for autophagy.

Overall the manuscript contains solid data that are backed up by adequate quantifications. The most novel contribution is the identification of an interaction between TBC1D14 and TRAPP and the evidence that this interaction is important for autophagy regulation. However, the mechanism for this regulation remains incompletely characterized.

We thank the reviewer for their positive and constructive comments and we have used these as a basis to improve our study.

Specific remarks:

1. A central conclusion from this study is that TBC1D14 serves to recruit TRAPPIII to the ERGIC to activate RAB1. However, this needs to be verified by studying RAB1 activation under conditions of TBC1D14 depletion and overexpression.

We agree with this comment and decided to address the issue of RAB1 activation under these conditions using a RAB1 effector pulldown approach. We based our experiments on the previous findings from the Lowe (Diao et al, JCB160:201-212, 2003) and Warren (Satoh et al., Traffic 4:153-161, 2003) groups who identified Golgin-84 as a Rab1 effector. These data are shown in new Figure EV2E and F. Comparing overexpression of myc-TBC1D14 to transfection of an empty vector, we observed no change in the amount of RAB1B bound to the RAB1 effector GFP-Golgin-84 (Figure EV2E). However, depletion of TBC1D14 results in a significant decrease in the relative amount of RAB1B bound to GFP-Golgin-84, comparable to depletion of the TRAPP subunits TRAPPC8 and TRAPPC4 (Figure EV2F). As stated in the revised manuscript, we propose that TBC1D14 is necessary but insufficient for TRAPPIII RAB1 GEF activity and that the disruption of autophagy and Golgi morphology on TBC1D14 overexpression is due to mis-localisation TRAPP complex to transferrin positive tubules by TBC1D14, rather than loss of TRAPP GEF activity.

2. The model in Fig. 9c proposes that Rab11 not only localizes to RE but also to ERGIC and cis-Golgi elements. However, the evidence is not convincing on this point. In Fig. 2A it is possible to observe a few structures that apparently contain both RAB11A and RAB1B in cells transfected with GFP-TBC1D14, but if such a co-localization can only be observed under GFP-TBC1D14 overexpression conditions its biological relevance remains uncertain. In addition, the resolution of the light microscope is insufficient to resolve distinct but juxtaposed vesicles, and immunoelectron microscopy would be required to conclude whether RAB11A and RAB1B truly co-localize on ERGIC and Golgi membranes.

To address whether RAB1 and RAB11 can colocalise in the absence of TBC1D14 overexpression, we carried our further careful confocal analysis of cells stained for RAB1, RAB11 and the golgi marker GM130. Previous studies have indicated that there is exchange between RAB11 and RAB1 positive compartments which is enhanced by BFA treatment (Marie et al, MBoC 20:4458-4470, 2009).

Based on this study we treated cells with Brefeldin A (BFA), an inhibitor of SEC7 domain proteins, and found that endogenous RAB11 and RAB1 co-localise on peripheral, stable punctate structures. Importantly we do not find GM130 on these structures, as GM130 localizes to ER exit sites in the presence of BFA (Mardones et al., MBoC 17:525-538, 2006). These data now form Figure EV1. We agree the best proof would be with immunogold labelling. However, despite several attempts with several different Rab1, Rab11, and ERGIC antibodies we were unable to optimise immunoelectron microscopy to localise endogenous RAB1, RAB11 and ERGIC markers. Given this result and the new confocal data shown in Figure EV1, we have modified our conclusions to propose that endogenous RAB11 and RAB1 transiently co-localise on a trafficking intermediate, and that TBC1D14 overexpression-induced tubules may contribute to this compartment.

3. The model in Fig. 9c implies that TRAPP III is a positive regulator of autophagy because it activates RAB1, a GTPase required for autophagy. This is confusing in light of previously published results from the Tooze lab that place TBC1D14 as an inhibitor of autophagy.

We agree with this reviewer and reviewer 1 that we did not integrate our previous findings with our new data. In light of our new data regarding RAB1 activation, and colocalisation of RAB1 and RAB11 on Brefeldin A treatment, we have clarified and modified our model. We propose that TBC1D14 may act to balance direct contribution of membrane to the autophagosome from the peripheral recycling endosome (via its RAB11 binding function) with ATG9 cycling (via its role in TRAPPIII function), and have included this discussion along with the improved model in Figure 9C. Thus, TBC1D14 acts as a negative regulator upon overexpression, or as an inhibitory effector of Rab11 in situations where active Rab1 and TRAPPIII may not yet be engaged. Whether, and how, this balance is regulated remains to be characterized, although the interplay of TBC1D14 with other recycling endosome localised autophagy regulators such as SNX18 (Knaevelsrud et al 2013, JCB) may be important in this regard.

2nd Editorial Decision

24 November 2015

Thank you for submitting your revised manuscript for our consideration.

Your manuscript has now been seen once more by the original referees (see comments below), and I am glad to inform you that they are both in favor of publication. I am therefore happy to inform you that I can accept your manuscript in principle for publication here.

REFeree COMMENTS

Referee #1:

This is an elegant and important study, which is now appropriately revised. I have no further comments.

Referee #2:

The authors have addressed the points I raised. I would like to recommend publication in EMBO Journal even though the revised model is a bit complicated.